# Terahertz radiation by quantum interference of excitons in a one-dimensional Mott insulator

Tatsuya Miyamoto [1]✉, Akihiro Kondo[1], Takeshi Inaba[1], Takeshi Morimoto[1], Shijia You[1] & Hiroshi Okamoto [1]✉

Nearly monocyclic terahertz waves are used for investigating elementary excitations and for controlling electronic states in solids. They are usually generated via second-order optical nonlinearity by injecting a femtosecond laser pulse into a nonlinear optical crystal. In this framework, however, it is difficult to control phase and frequency of terahertz waves. Here, we show that in a one-dimensional Mott insulator of a nickel-bromine chain compound a terahertz wave is generated with high efficiency via strong electron modulations due to quantum interference between odd-parity and even-parity excitons produced by two-color femtosecond pulses. Using this method, one can control all of the phase, frequency, and amplitude of terahertz waves by adjusting the creation-time difference of two excitons with attosecond accuracy. This approach enables to evaluate the phase-relaxation time of excitons under strong electron correlations in Mott insulators. Moreover, phase- and frequency-controlled terahertz pulses are beneficial for coherent electronic-state controls with nearly monocyclic terahertz waves.

Recent developments in femtosecond laser technology have enabled the generation of nearly monocyclic terahertz pulses with electric-field amplitudes far exceeding 100 kV/cm[1–4], which originated from optical rectification (OR) in second-order nonlinear optical crystals excited with a visible femtosecond pulse. The intense terahertz pulse has been used as an excitation pulse to control the electronic state properties such as electric conductivity[5–8], superconductivity[9–11], dielectricity[12–18], magnetism[19–23], and topology[24]. Terahertz pulses with variable phase, frequency and amplitude are useful for controlling the electronic states of matter[25–28] but are challenging to achieve in the simple OR framework in which a second-order nonlinear optical crystal is excited with a femtosecond laser pulse in the transparent region. An effective method for this is to use an OR process based upon the third-order optical nonlinearity associated with two pulses with the frequencies of $\omega$ and $2\omega$, which is expressed by $P(0) = \frac{3}{4}\varepsilon_0\chi^{(3)}(0;\omega,\omega,-2\omega)E(\omega)E(\omega)E(-2\omega)$. Here, $E(\omega)$ and $E(-2\omega)$ are the electric fields of $\omega$- and $2\omega$-pulses, respectively. This process causes a finite photocurrent even in centrosymmetric materials[29–31]

and therefore may extend the possibility to generate terahertz electromagnetic waves effectively. In fact, in conventional semiconductors, i.e., IV semiconductors of silicon (Si) and germanium (Ge), III-V semiconductors of GaAs and InP, and II-VI semiconductors of CdSe and CdTe, terahertz radiations due to this mechanism were detected[32–34]. In these phenomena, photo-excitations fundamentally generate interband transitions so that the dynamics of photoexcited carriers under the influence of their complicated scattering processes with other carriers and phonons would affect the temporal behaviors of induced currents. This makes it difficult to clarify the detailed terahertz radiation mechanism, and possibly suppresses terahertz radiation efficiencies. In II-VI semiconductors, the possibility of controlling the phase of terahertz radiation using a transition between different excitonic states has also been investigated[34]. However, due to the weak excitonic effect, the terahertz radiation associated with the excitonic transitions is much smaller than that originating from the interband transitions. We expect that more advanced controls of terahertz radiations in the framework of the third-order optical nonlinearity,

[1]Department of Advanced Materials Science, University of Tokyo, Chiba 277-8561, Japan. ✉e-mail: miyamoto@edu.k.u-tokyo.ac.jp; okamotoh@k.u-tokyo.ac.jp

namely the generations of phase-, frequency-, and amplitude-adjustable terahertz pulses with high efficiency would be possible by utilizing specific materials having well-defined wavefunctions of photoexcited states and large values of related $\chi^{(3)}(0;\omega,\omega,-2\omega)$.

On the basis of these backgrounds, in this study, we experimentally demonstrate an efficient generation method of nearly monocyclic terahertz pulses with adjustable phase, frequency, and amplitude using excitonic states in one-dimensional (1D) Mott insulators, the linear and nonlinear optical properties of which have been actively studied in the community of condensed matter physics[35–46]. We focus a half-filled 1D electronic system with one orbital occupied by one electron. When the on-site Coulomb repulsion $U$ is larger than the transfer integral $t$ between neighboring orbitals, the system becomes a Mott insulator. In our study, we chose a bromine-bridged nickel-chain compound, [Ni(chxn)$_2$Br]Br$_2$ (chxn = cyclohexanediamine), as a target material. In this compound, Ni$^{3+}$ and Br$^-$ ions are arranged alternately along the $b$-axis (Fig. 1a)[47]. The singly occupied Ni-$3d_{z^2}$ orbitals, which are hybridized via the Br-$4p_z$ orbitals, form a 1D electronic band. The Ni-$3d$ band is split into the upper Hubbard band (UHB) and lower Hubbard band (LHB) due to the large $U$ in the Ni-$3d_{z^2}$ orbital. Moreover, the Br-$4p$ band is located between the UHB and LHB, and the charge-transfer (CT) transition from the Br-$4p$ band to Ni-$3d$ UHB corresponds to the optical gap (Fig. 1b). Although this material is strictly a CT insulator described by the two-band Hubbard model considering Br-$4p$ and Ni-$3d_{z^2}$ orbitals[39,48], the single-band Hubbard model can explain the fundamental electronic properties of this material qualitatively[36,38,40,41]. For this reason, we will simply call it a 1D Mott insulator in this paper.

In 1D Mott insulators, when the nearest-neighbor Coulomb repulsion energy is large, the low-energy electronic excited state is a doublon-holon bound state, i.e., an exciton[49]. In this case, the lowest $|\varphi_o\rangle$ and second-lowest $|\varphi_e\rangle$ excitons are out of phase and in phase combination with an electron excitation from a Br ion to the left and right Ni ion ($|L\rangle$ and $|R\rangle$) (Fig. 1b), expressed as $|\varphi_o\rangle = \frac{(|L\rangle - |R\rangle)}{\sqrt{2}}$ and $|\varphi_e\rangle = \frac{(|L\rangle + |R\rangle)}{\sqrt{2}}$, respectively. Thus, $|\varphi_o\rangle$ and $|\varphi_e\rangle$ are the odd-parity and even-parity excitons, respectively. If the hole in an exciton is located at the $i$ site, the envelopes of the electron wavefunctions of $|\varphi_o\rangle$ and $|\varphi_e\rangle$ are as indicated by the blue and red lines in Fig. 1c, d, respectively. In this case, their wavefunctions have similar spatial extensions except for their phases (Fig. 1c, d). Thus, their energy difference becomes minimal, and their transition dipole moment, $\langle\varphi_o|x|\varphi_e\rangle$, is enhanced[35,36,40]. Owing to the large $\langle\varphi_o|x|\varphi_e\rangle$, 1D Mott insulators exhibit large

third-order optical nonlinearity such as efficient third-harmonic generations (THG)[38,41] and two-photon absorptions[37,43], and large electric-field changes of reflectivity[35,40,46]. These features of excited states and large third-order optical nonlinearity in 1D Mott insulators were first explained by the single-band Hubbard model. Subsequently, calculations using the two-band Hubbard model have revealed that CT insulators indeed have similar features[39,42,44,45]. It is expected that this novel third-order optical nonlinearity associated with nearly degenerate odd and even parity excitons, $|\varphi_o\rangle$ and $|\varphi_e\rangle$, in 1D Mott insulators can be used to produce strong terahertz pulses with freely adjustable phase.

The previous theoretical study focusing on the compound semiconductors predicted that charge current would be generated when two excitons with different symmetry are simultaneously excited and that the phase of the current could be changed by varying the phase difference between the polarizations of two excitons[50]. In this case, the terahertz radiation and its phase control might be possible. In the following, we show the formulation of the terahertz radiation from the transition between odd- and even-parity excitons photogenerated in a 1D Mott insulator. Let us consider the case that two types of excitons, $|\varphi_o\rangle$ and $|\varphi_e\rangle$, are produced at time $t = T$ and $t = 0$, respectively, using two femtosecond laser pulses with different photon energies. In this case, the wavefunction of the system, $|\psi\rangle$, becomes as follows.

$$|\psi\rangle = A\exp(-i\omega_o(t-T))|\varphi_o\rangle + B\exp(-i\omega_e t)|\varphi_e\rangle \\ + \sqrt{1 - |A|^2 - |B|^2}|\varphi_g\rangle \tag{1}$$

In Eq. (1), $A(B)$ and $\omega_o(\omega_e)$ are the amplitude of the wavefunction and frequency of the odd parity exciton $|\varphi_o\rangle$ (the even-parity exciton $|\varphi_e\rangle$), respectively, and $|\varphi_g\rangle$ is the original ground state. The time evolution of the real part of the system's wavefunction, Re$|\psi\rangle$, at the $(i + 1)$ and $(i - 1)$ sites at $T = 0$ is represented by the black lines in Fig. 1d. At the $(i + 1)$ site, the initial signs of the amplitudes of the wavefunctions $|\varphi_o\rangle$ and $|\varphi_e\rangle$ are the same, and they interfere constructively at $t = 0$. However, at the $(i - 1)$ site, their initial signs are different, and interfere destructively. Subsequently, the envelope of Re$|\psi\rangle$ (each black line) oscillates at the differential frequency of the odd-parity and even-parity excitons, $(\omega_e - \omega_o)$. Therefore, the average electron density at the $(i + 1)$ and $(i - 1)$ sites oscillates with a frequency of $(\omega_e - \omega_o)$, as shown by the green line in Fig. 1d. This electron oscillation corresponds to the low-frequency component $P_L$ in the expected

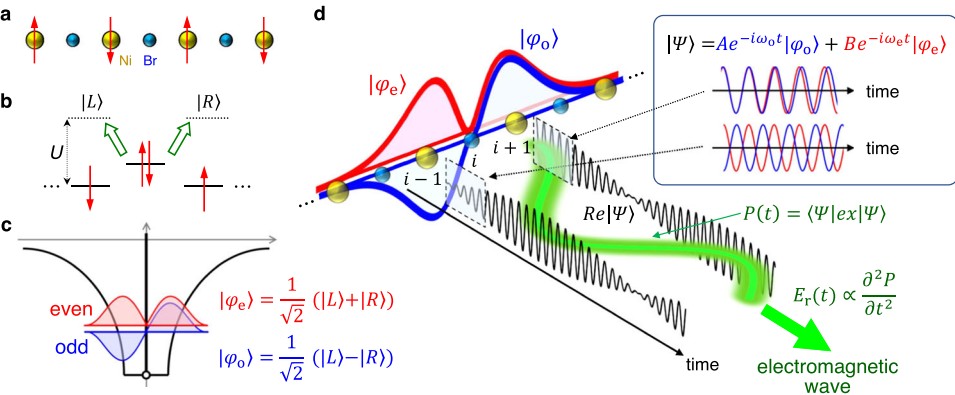

**Fig. 1 | 1D chain structure in [Ni(chxn)$_2$Br]Br$_2$, schematics of excitonic states and possible radiation from an interference of exciton wavefunctions in a 1D Mott insulator. a** 1D Ni-Br chain. Red arrows show electron spins. **b** Schematic of electron excitations from a Br ion to the left and right Ni ion ($|L\rangle$ and $|R\rangle$). $U$ is the on-site Coulomb repulsion. **c** Schematic of electron wavefunctions of the odd-parity exciton $|\varphi_o\rangle$ and the even-parity exciton $|\varphi_e\rangle$. **d** Time evolutions of the real

part of the wavefunction of the system, Re$|\psi\rangle$, at $(i+1)$ and $(i-1)$ sites (the black lines). The green curve shows an expected oscillation of the averaged electron density with the frequency of $(\omega_e - \omega_o)$ corresponding to the low-frequency component in the expected value of the polarization, $P = e\langle\psi|x|\psi\rangle$, from which an electromagnetic wave with the electric field $E_r \left(\propto \frac{\partial^2 P}{\partial t^2}\right)$ is emitted.

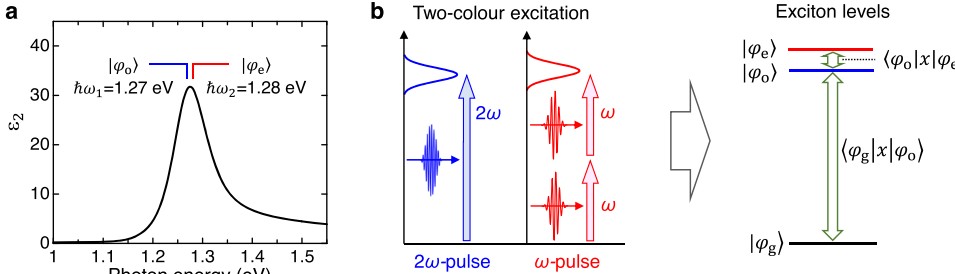

**Fig. 2 | Exciton spectrum and two-color excitation of excitons in [Ni(chxn)$_2$Br] Br$_2$. a** The spectrum of imaginary part of the dielectric constant, $\varepsilon_2$, along the 1D Ni-Br chains, in which the energies of odd-parity and even-parity excitons ($\hbar\omega_1 = 1.27$ eV and $\hbar\omega_2 = 1.28$ eV) are indicated. **b** Schematic diagram of the two-color excitation of the odd-parity exciton $|\varphi_o\rangle$ with the frequency of $2\omega$ and the even-parity exciton $|\varphi_e\rangle$ with the frequency of $\omega$. $|\varphi_g\rangle$ represents the ground state. $\langle\varphi_g|x|\varphi_o\rangle$ ($\langle\varphi_o|x|\varphi_e\rangle$) are the transition dipole moments between $|\varphi_g\rangle$ and $|\varphi_o\rangle$ ($|\varphi_o\rangle$ and $|\varphi_e\rangle$).

value of the polarization, $P = e\langle\psi|x|\psi\rangle$, expressed as

$$P_L = eA^*Be^{-i(\omega_e - \omega_o)t - i\omega_o T}\langle\varphi_o|x|\varphi_e\rangle + \text{c.c.} \quad (2)$$

Consequently, an electromagnetic wave with an electric field $E_r \propto \frac{\partial^2 P_L}{\partial t^2}$ is emitted. To efficiently generate terahertz waves via this process, ($\omega_e - \omega_o$) must be in the terahertz region. Figure 2a shows the linear absorption spectrum, that is, the imaginary part of the dielectric constant, $\varepsilon_2$, along the *b*-axis. It has a peak structure at 1.27 eV, which is attributed to the odd-parity exciton. Previous electro-reflectance spectroscopy results and THG studies revealed that the even-parity exciton is located at an energy level 10 meV (~2.5 THz) above the odd-parity exciton (Fig. 2a)[40]. This satisfies the condition that ($\omega_e - \omega_o$) is in the terahertz region. Under this condition, $P_L$ is expected to be enhanced owing to the large $\langle\varphi_o|x|\varphi_e\rangle$. By controlling the quantum interference of the two wavefunctions of the odd-parity and even-parity excitons with attosecond accuracy[51–53], the phase of $P_L$ or the generated terahertz waves, $-i\omega_o T$, can be continuously varied. Note that other methods for the efficient generation of terahertz electromagnetic wave via the third-order optical nonlinearity in 1D Mott insulators have been theoretically proposed[42,44,45], while in those methods the phase of the terahertz electromagnetic wave cannot be controlled unlike the method proposed in the present study.

In this work, we demonstrate that terahertz electromagnetic waves are generated via the quantum interference between the odd-parity and even-parity excitons in [Ni(chxn)$_2$Br]Br$_2$, which are excited via one-photon and two-photon absorption processes, respectively, using two-color femtosecond laser pulses. The amplitude and phase of radiated terahertz waves can be controlled independently; by setting the appropriate value of the interval between the incident $\omega$- and $2\omega$-pulses, that is, the creation time difference of two excitons with the accuracy of ten attoseconds, we can select the continuous amplitude control of the terahertz radiation with its phase fixed to $\pm\pi$ or the continuous phase control from $-\pi$ to $+\pi$ with its amplitude fixed. We also demonstrate that the central frequency of the terahertz radiation can be tuned by adjusting the creation-time difference of two excitons. Moreover, we show that the efficiency of the terahertz radiation in [Ni(chxn)$_2$Br]Br$_2$ is much higher than those in conventional semiconductors, Si and Ge, in the similar framework of the third-order optical nonlinearity. The efficiency of the terahertz radiation in [Ni(chxn)$_2$Br]Br$_2$ is also higher than that from ZnTe originating from the general OR framework based upon the second-order optical nonlinearity. In the final part, we discuss the phase relaxation dynamics of excitons in [Ni(chxn)$_2$Br]Br$_2$ by analyzing the dependence of the terahertz radiation intensity on the creation time difference of even-parity and odd-parity excitons.

## Results

### Terahertz radiation under resonant excitation conditions

First, we report the terahertz radiation produced when odd- and even-parity excitons were simultaneously excited. The even-parity exciton was excited via two-photon absorption of a pulse with $\hbar\omega = 0.64$ eV (Fig. 2b), which is hereafter called the $\omega$-pulse. The odd-parity exciton was excited via a linear (one-photon) absorption of the second harmonic ($2\hbar\omega = 1.28$ eV) of the $\omega$-pulse, called the $2\omega$-pulse. The delay time between the $2\omega$-pulse and $\omega$-pulse is $\Delta t$. These pulses have spectral widths of approximately 20 meV, and the odd-parity and even-parity excitons have widths of approximately 50 meV and 130 meV, respectively[46]. Therefore, these pulses can generate both odd-parity and even-parity excitons. The experimental details are reported in the Methods section and Supplementary Notes 1 and 2.

Figure 3b shows the electric-field waveforms of the terahertz radiation, $E_{THz}(t)$, when the two pulses are incident with a small $\Delta t$ ranging from $-2.08$ fs to 1.19 fs. Because the temporal width of each pulse is ~100 fs, the two pulses are incident on the sample almost simultaneously as shown in Fig. 3a. However, the waveform of $E_{THz}(t)$ is sensitive to $\Delta t$ over the time scale of 10 attoseconds. Here, we express $E_{THz}(t)$ as $E_{THz0}(t)\cos(\omega_{THz}t - \phi)$, where $E_{THz0}(t)$ and $\phi$ are the envelope and phase of $E_{THz}(t)$, respectively. $\phi$ was evaluated from the phase of a 1.0 THz wave using the Fourier transformation of the terahertz waveforms. At $\Delta t = -2.08$ fs, $\phi$ is zero. As $\Delta t$ increases to $-1.76$ fs, the amplitude of the terahertz wave decreases as its phase $\phi$ remains constant. Further increasing $\Delta t$ changes $\phi$ from 0 to $\pi$ near $-1.7$ fs, and the absolute amplitude increases conversely and is maximized near $-0.8$ fs. A similar change occurs over time: $\phi$ alternates between 0 and $\pi$ with a period of 3.24 fs (Fig. 4a), which corresponds to a photon energy of 1.28 eV. This implies that $E_{THz0}$ is proportional to $\cos(2\omega\Delta t)$.

To investigate the nonlinearity of the terahertz radiation, we measured the dependence of the amplitudes of emitted terahertz waves on the incident electric field. The results are reported in Supplementary Note 3. The amplitudes at the time origin, $E_{THz}(0)$, with $\Delta t = 0.86$ fs are plotted as a function of the product of the square of the electric field amplitude of the $\omega$-pulse ($E_\omega$) and the $2\omega$-pulse ($E_{2\omega}$), $E_\omega E_\omega E_{2\omega}$ in Supplementary Fig. 5. $E_{THz}(0)$ is proportional to $E_\omega E_\omega E_{2\omega}$, indicating that the observed response is attributable to third-order optical nonlinearity called four-wave rectification[32–34,54].

Increasing $\Delta t$ to approximately 60 fs (Fig. 3c) varies the electric field waveforms of the terahertz radiation considerably as shown in Fig. 3d. In contrast to the case of $\Delta t \sim 0$ fs (Fig. 3b), the phase $\phi$ of $E_{THz}(t)$ changes continuously with the increase of $\Delta t$, whereas the amplitude is independent of $\Delta t$. Moreover, when $\phi$ is defined in the range of $|\phi| \leq \pi$, it decreases from $\pi$ to $-\pi$ with increasing $\Delta t$ as shown in Fig. 4b. Namely, $\phi$ can be freely changed by adjusting the $\Delta t$ value. Thus, by selecting the appropriate value of $\Delta t$, we can achieve two types of controls of amplitude and phase of terahertz radiations; the

continuous control of the electric-field amplitude with the electric-field phase fixed to $\pm\pi$ or the continuous control of the electric-field phase from $-\pi$ to $+\pi$ with the electric-field amplitude fixed.

To investigate the observed $\Delta t$ dependence of the terahertz waves on the resonant exciton excitation (Fig. 5a) in more detail, we calculated the Fourier-amplitude spectra based on $E_{THz}(t)$, and the normalized results are shown in Fig. 5b. The spectrum peaks at approximately 1 THz at $\Delta t = 0$ fs and shifts to a higher frequency with increasing $\Delta t$. This indicates that the frequency of the terahertz wave can also be controlled by adjusting $\Delta t$. The reason of this frequency shift of the terahertz wave is discussed in Discussion section.

We next integrated each spectrum from 0 to 3 THz and plotted the intensity $I_{THz}(\Delta t)$ against $\Delta t$, which is shown in Fig. 5c: a prominent oscillation with a period of 1.62 fs is observed especially at $\Delta t \sim 0$fs. $I_{THz}(\Delta t)$ at $\Delta t \sim 0$fs is approximately reproduced by $|\cos(2\omega\Delta t)|$ as shown by the gray line in the inset of Fig. 5c. This suggests that $E_{THz}(0)$ includes components that are proportional to $\cos(2\omega\Delta t)$. As $\Delta t$ increases, the oscillation amplitude decreases and becomes very small at $\Delta t > 60$ fs. In contrast, the non-oscillating background component gradually increased, peaked at $\Delta t \sim 25$ fs, and then decreased. The oscillatory and background components are attributable to a coherent response driven by the electric fields of the two incident pulses and a response from the generated excitons, respectively. Hereafter, the latter is called the real exciton response.

## Terahertz radiation under non-resonant excitation conditions

To elucidate the contribution of the coherent response, we next measured the terahertz radiation under non-resonant excitation conditions (Fig. 5d), where a 0.585 eV $\omega$-pulse and 1.17 eV $2\omega$-pulse were used. The obtained terahertz electric-field waveforms are shown in Supplementary Note 3 and their Fourier-amplitude spectra are presented in Fig. 5e, both of which are independent of $\Delta t$ and similar to those obtained at $\Delta t \sim 0$ fs in the resonant exciton excitation shown in Figs. 3b and 5b, respectively. The Fourier-amplitude integrated from 0 to 3 THz, $I_{THz}(\Delta t)$, is shown in Fig. 5f. $I_{THz}(\Delta t)$ depends on $\Delta t$ as $|\cos(2\omega\Delta t)|$, as shown by the gray line in the inset of Fig. 5f. This behavior is similar to that observed at $\Delta t \sim 0$ fs in the resonant exciton excitation. Thus, the response defined by $\cos(2\omega\Delta t)$ can be regarded as the coherent response.

## Comparison of terahertz radiations from [Ni(chxn)$_2$Br]Br$_2$ and conventional semiconductors

As mentioned in the introductory part, in conventional semiconductors with inversion symmetry such as Si, Ge, InP, and GaAs, terahertz radiations based upon the third-order nonlinear optical process expressed by $P(0) = \frac{3}{4}\varepsilon_0\chi^{(3)}(0;\omega,\omega,-2\omega)E(\omega)E(\omega)E(-2\omega)$ using $\omega$- and $2\omega$-pulses have also been reported. In those semiconductors, the two-photon excitation by the $\omega$- pulse and one-photon excitation by the $2\omega$-pulse are both due to the interband transitions. It is interesting to compare the efficiency of the terahertz radiation from

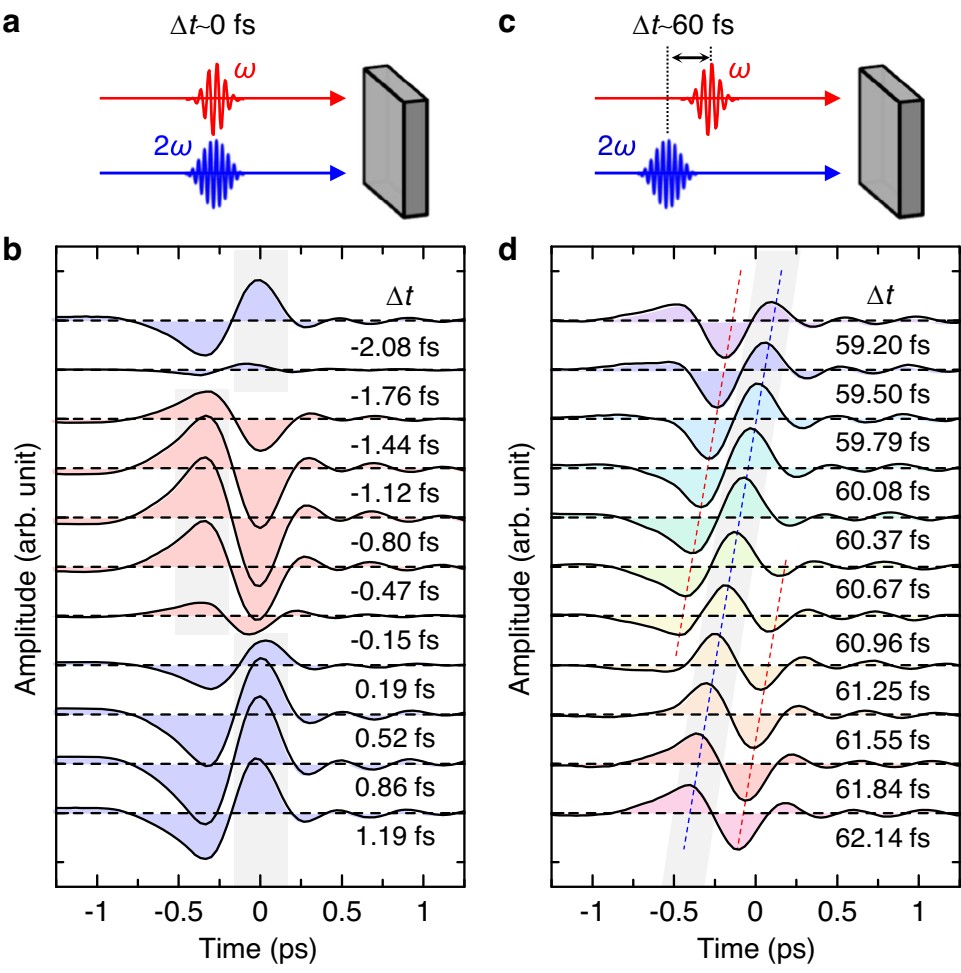

**Fig. 3 | Terahertz radiation by the resonant excitation of even- and odd-parity excitons with $\omega$- and $2\omega$-pulses in [Ni(chxn)$_2$Br]Br$_2$.** Schematic diagram of the $\omega$- and $2\omega$-pulse injection with the delay time (**a**) $\Delta t \sim 0$ fs and (**c**) $\Delta t \sim 60$ fs on to a crystal. Electric-field waveforms of terahertz radiations at (**b**) $\Delta t \sim 0$ fs and (**d**) $\Delta t \sim 60$ fs.

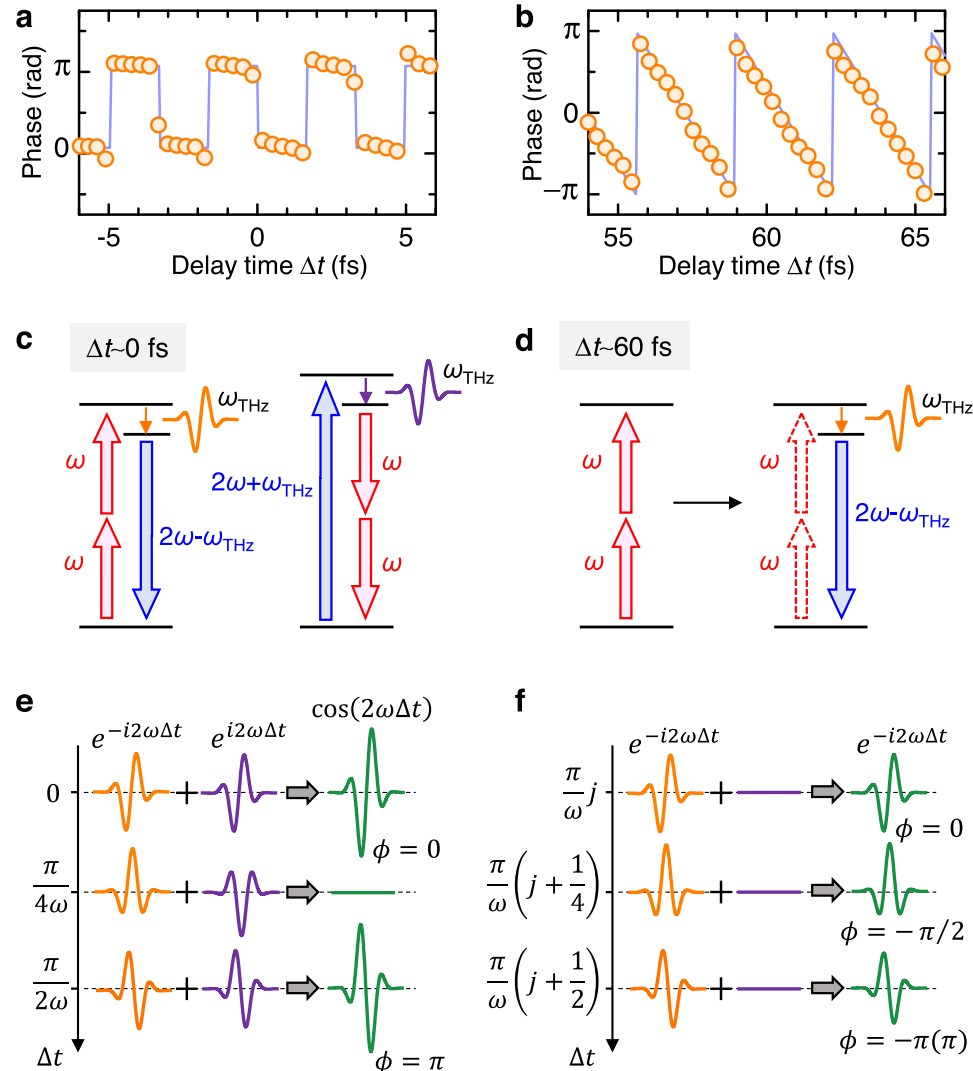

**Fig. 4 | Phase of terahertz electric fields depending on the creation time difference between even- and odd-parity excitons in [Ni(chxn)₂Br]Br₂.** The phase of the electric-field waveform as a function of $\Delta t$ at (**a**) $\Delta t \sim 0$ fs and (**b**) $\Delta t \sim 60$ fs. The situations with $\phi = \pi$ and $\phi = -\pi$ are equivalent and the phase $\phi$ is determined in the range of $-\pi < \phi \leq \pi$. Schematic diagrams of terahertz radiation processes with the frequency of $\omega_{THz}$; (**c**) the simultaneous injection of $\omega$- and $2\omega$-pulses and (**d**) the injection of $\omega$- and $2\omega$-pulses with $\Delta t \sim 60$ fs. Schematic diagram of the interference of two types of four-photon processes shown in (**c**); (**e**) $\Delta t \sim 0$ fs and (**f**) $\Delta t \sim 60$ fs. $j$ is a natural number. At $\Delta t \sim 60$ fs, only the process shown in d is dominant and no interference occurs.

[Ni(chxn)₂Br]Br₂ with those from those semiconductors, which originate from the similar third-order nonlinear optical processes. For this subject, we have measured the terahertz radiations from [Ni(chxn)₂Br]Br₂, Si, and Ge with two incident pulses of $\hbar\omega = 0.64$ eV and $2\hbar\omega = 1.28$ eV in the same experimental setup with the reflection configuration (see Supplementary Note 1). For Si and Ge, crystals cut out of the (100) plane were used. The excitation fluences of the $\omega$- and $2\omega$-pulses are 1.77 mJ/cm² and 0.11 mJ/cm², respectively. The $\omega$-pulse was used for the two-photon excitation, and its fluence was much higher than that of the $2\omega$-pulse used for the one-photon excitation.

In Fig. 6, the electric field waveforms of the terahertz radiations from Si and Ge are shown by the green and red lines, respectively, together with that emitted from [Ni(chxn)₂Br]Br₂ (the black line). The electric field amplitude (intensity) of the terahertz radiation from [Ni(chxn)₂Br]Br₂ is approximately 2.4 (5.8) times higher than that from Ge and 50 (2500) times higher than that from Si. The terahertz radiation from Si is much smaller than that from Ge. It is because the direct interband transition cannot be excited by the 1.28 eV pulse in Si but can be excited in Ge.

Next, we compare the terahertz radiation efficiency of [Ni(chxn)₂Br]Br₂ and Ge more precisely. In our experiments, the polarizations of $\omega$- and $2\omega$-pulses are orthogonal (see Supplementary Fig. 1b). In the experiments on [Ni(chxn)₂Br]Br₂, they are tilted by ±45 degrees with respect to the 1D Ni chain (see Supplementary Note 1). If both polarizations are parallel to the 1D Ni chain (the $b$-axis) while maintaining the intensities of two pulses, the electric field amplitude of the terahertz radiation is expected to increase by a factor of $2\sqrt{2}$ and its intensity by a factor of 8. On the other hand, in Ge, the terahertz radiation intensity for the polarizations of two excitation pulses parallel to each other is roughly 1.1 times higher than that for their polarizations perpendicular to each other[33]. Therefore, the intensity of the terahertz radiation from [Ni(chxn)₂Br]Br₂ is estimated to be approximately 42 times (= 5.8 × 8/1.1) as large as that from Ge. When the terahertz radiation generated in this way is actually used to excite materials, the damage of each crystal when the excitation fluence is increased must also be considered. In the present experiment, when $\omega$- and $2\omega$-pulses were irradiated simultaneously, the thresholds of the fluences of $\omega$- and $2\omega$-pulses at which they cause damage of the crystal

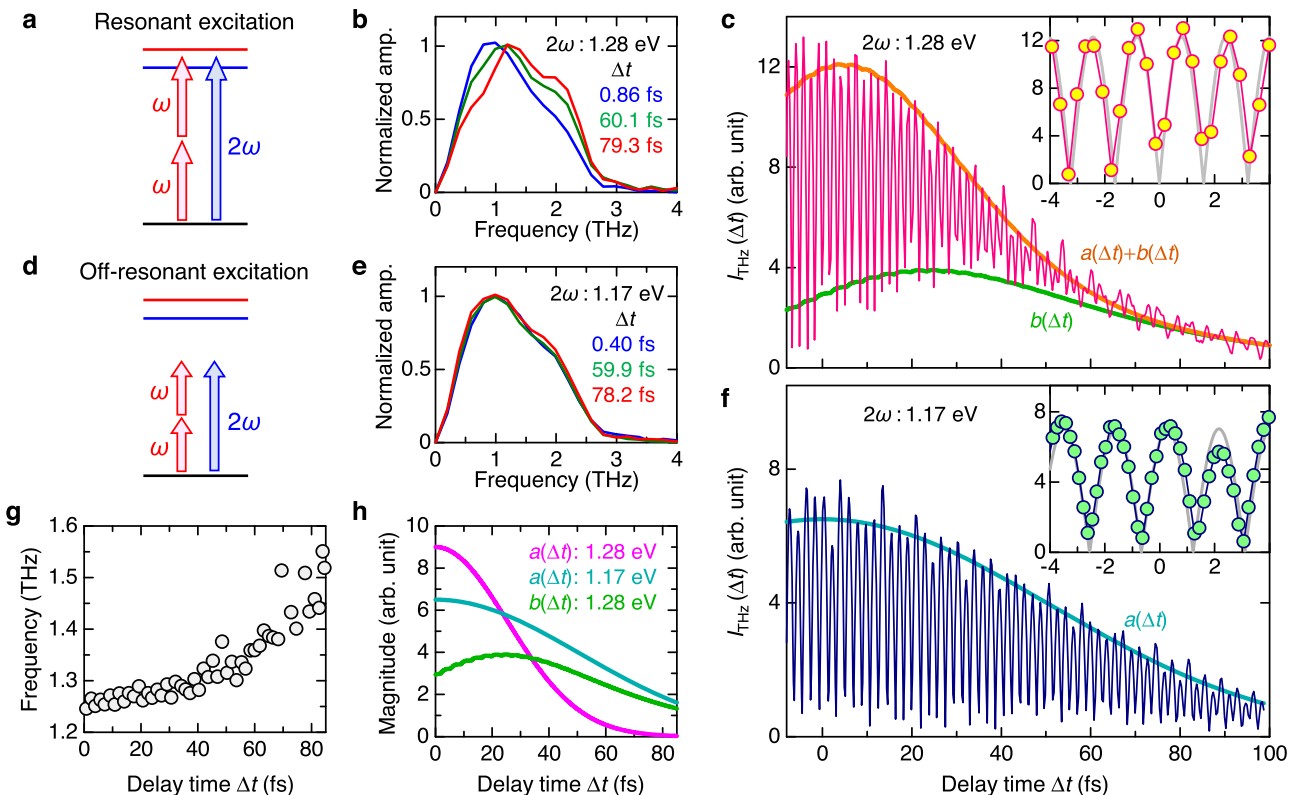

**Fig. 5 | Spectral features of terahertz radiations by resonant and non-resonant excitation of even-parity and odd-parity excitons in [Ni(chxn)₂Br]Br₂.**
**a** Schematic diagram of resonant excitation of even- and odd-parity excitons with $\omega$- and $2\omega$-pulses, respectively. **b** Typical spectra of electric-field amplitudes of the terahertz radiations for the resonant excitation with $2\hbar\omega = 1.28$ eV ($\hbar\omega = 0.64$ eV). **c** $\Delta t$ dependence of integrated electric-field amplitude of the terahertz radiation, $I_{\mathrm{THz}}(\Delta t)$, for the resonant excitation. The orange and green lines show the simulation curves of the envelope, $[a(\Delta t) + b(\Delta t)]$, and the non-oscillating component, $b(\Delta t)$, respectively. The gray line in the inset is the simulation curve at $\Delta t - 0$ fs. **d** Schematic diagram of non-resonant excitation. **e** Typical spectra of electric-field amplitudes of the terahertz radiations for the non-resonant excitation with $2\hbar\omega = 1.17$ eV ($\hbar\omega = 0.585$ eV). **f** $\Delta t$ dependence of $I_{\mathrm{THz}}(\Delta t)$ for the non-resonant excitation. The blue line shows the simulation curve of the envelope, $a(\Delta t)$. The gray line in the inset is the simulation curve at $\Delta t - 0$ fs. **g** $\Delta t$ dependence of spectral center of gravity of the electric-field amplitude spectrum. **h** $\Delta t$ dependence of $a(\Delta t)$ and $b(\Delta t)$ reflecting the coherent and real exciton responses, respectively.

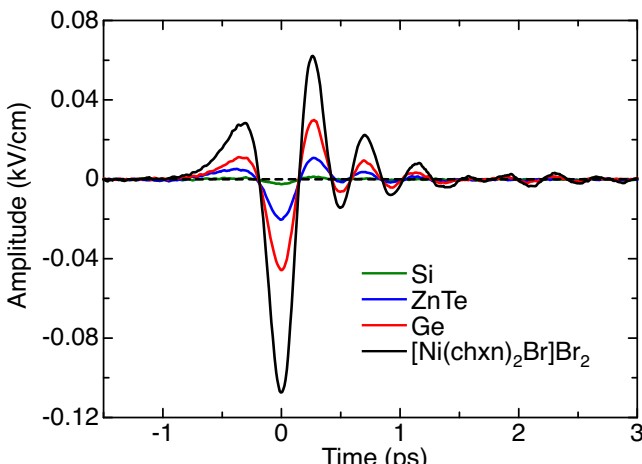

**Fig. 6 | Comparison of the electric field waveforms of terahertz radiations between [Ni(chxn)₂Br]Br₂ and other semiconductors.** The green, blue, red and black solid lines are the electric field waveforms of terahertz radiations from Si, ZnTe, Ge and [Ni(chxn)₂Br]Br₂, respectively.

were 11.7 mJ/cm² and 0.86 mJ/cm² in [Ni(chxn)₂Br]Br₂ (23.4 mJ/cm² and 1.74 mJ/cm² in Ge), respectively. The terahertz electric-field amplitude tends to saturate when the pulse fluences approach the damage thresholds in both materials. Just below the damage threshold, the

terahertz electric-field amplitude was found to be 0.53 kV/cm in [Ni(chxn)₂Br]Br₂ and 0.23 kV/cm in Ge. Considering these values, the ratio of their radiation intensities is decreased from 42 to 39, but the superiority of [Ni(chxn)₂Br]Br₂ remains unchanged.

More strictly, it is necessary to discuss the terahertz radiation efficiency by taking the penetration depths of the incident pulses into account. Since the penetration depth of the $2\omega$-pulse, $l(2\omega)$, is much shorter than that of the $\omega$-pulse, $l(\omega)$, the terahertz radiation is emitted only from a region as deep as $l(2\omega)$. Therefore, it is reasonable to consider that an electric-field amplitude of the terahertz radiation is proportional to $l(2\omega)$ and its intensity is proportional to $[l(2\omega)]^2$. $l(2\omega)$ is approximately 20 nm in [Ni(chxn)₂Br]Br₂ and approximately 500 nm in Ge[55]. To compare the terahertz radiation efficiency per unit length between [Ni(chxn)₂Br]Br₂ and Ge, it is necessary to multiply the terahertz radiation intensity in the former approximately 600 times. Taking into account this factor and the intensity ratio of 42 estimated above, the terahertz radiation efficiency per unit length in [Ni(chxn)₂Br]Br₂ is estimated to be $2.6 \times 10^4$ times as large as that of Ge. This extremely high efficiency of the terahertz radiation in [Ni(chxn)₂Br]Br₂ is characteristic of 1D Mott insulators, in which third-order nonlinear optical responses associated with nearly degenerate odd- and even-parity excitons are very large. Such large third-order nonlinear optical responses originate from the effects of strong electron correlation and one-dimensionality[35,36,39,40].

In CdSe and CdTe, it has been attempted to generate the phase-controlled terahertz pulses by the excitations of 1s and 2p excitons with different symmetries[34]. However, the exciton binding energies of

these compound semiconductors are still small; the binding energy of the lowest exciton is approximately 15 meV in CdSe and 11 meV in CdTe. In those compounds, therefore, it is difficult to resonantly excite only those excitons and the generated terahertz waves necessarily include two types of the components, which originate from the interband transitions and from the exciton transitions. As a result, when varying the phase difference between the $\omega$- and $2\omega$-pulses, both amplitude and phase of terahertz radiation are simultaneously changed, and the continuous control of the electric-field phase of the terahertz radiation with the electric-field amplitude fixed, as demonstrated in [Ni(chxn)$_2$Br]Br$_2$, had not been achieved[32,34]. In addition, these compounds are three-dimensional (3D) semiconductors and oscillator strength of the lowest exciton is not so large. Therefore, the intensity of the terahertz radiation caused by the exciton transitions is much smaller than that caused by the interband transitions. In our study, in [Ni(chxn)$_2$Br]Br$_2$, the terahertz radiation occurs only via the quantum interference between the odd-parity and even-parity excitons and is not affected by the interband transitions. It is because the binding energies of the odd-parity exciton and even-parity exciton, which are approximately 90 meV and 80 meV, respectively[41], are sufficiently large as compared to the spectral width of the excitation pulses (approximately 20 meV) and the oscillation strength is concentrated on the lowest odd-parity exciton reflecting the 1D nature of the electronic system.

Using the same experimental setup, we also measured the terahertz radiation from ZnTe, which is a second-order nonlinear optical crystal without inversion symmetry and widely used to generate terahertz electromagnetic waves via the OR process described by $P(0) = \varepsilon_0 \chi^{(3)}(0; \omega, -\omega)E(\omega)E(-\omega)$ [56,57]. In general, the third-order nonlinear optical effect described by $P(0) = \frac{3}{4}\varepsilon_0 \chi^{(3)}(0; \omega, \omega, -2\omega)E(\omega)E(\omega)E(-2\omega)$ should be much smaller than the second-order nonlinear optical effect. Since the excitation fluence of the $\omega$-pulse is approximately 16 times higher than that of the $2\omega$-pulse, it can be considered that the terahertz radiation occurs via the OR induced by the $\omega$-pulse in ZnTe. The electric field waveform of the terahertz radiation from ZnTe with the (110) plane cut out is shown by the blue line in Fig. 6. The electric field amplitude of the terahertz radiation emitted from [Ni(chxn)$_2$Br]Br$_2$ is 5.4 times as large as that from ZnTe. Although the difference is that the former used resonant exciton excitations and the latter used non-resonant excitations in the transparent region, this result also indicates that the efficiency of the terahertz radiation from [Ni(chxn)$_2$Br]Br$_2$ is very high.

Controlling the phase and frequency of a terahertz pulse might be possible using a second-order nonlinear optical effect. In fact, it was reported that a mid-infrared pulse can be generated from two near-infrared pulses with different frequencies via the differential frequency generation (DFG) process[25,58,59]. In those studies, the tuning of phase and frequency of the mid-infrared pulse was achieved by adjusting the time difference between two near-infrared pulses. However, this technique would be somewhat difficult to apply for a terahertz radiation. It is because a terahertz radiation due to the simple single-pulse OR process occurs by each incident pulse, making it difficult to control the phase and frequency of the terahertz radiation via the DFG process. In addition, in the DFG method, a complicated measurement system was necessary to prepare two optical paths with precisely adjusted lengths for two different near-infrared pulses. In contrast, the method using the third-order optical nonlinearity reported here requires only two pulses with frequencies of $\omega$ and $2\omega$. The latter pulse can be easily obtained from the former pulse using the SHG process within the same optical path (see Supplementary Fig. 1a), so the measurement system is much more simplified.

Finally, we estimate the energy conversion efficiency from two excitation pulses to a terahertz pulse in [Ni(chxn)$_2$Br]Br$_2$ with our

method using the third-order optical nonlinearity. In our experiments, the total energy of the $\omega$- and $2\omega$-pulses incident to the sample is $1.8 \times 10^{-6}$ J, while the energy of the terahertz radiation emitted from [Ni(chxn)$_2$Br]Br$_2$ was evaluated to be $1.3 \times 10^{-13}$ J from the electric-field waveform $E_{THz}(t)$ quantitatively measured (see Methods section). Considering the correction factor of 8 mentioned above, we obtained the energy efficiency of $5.8 \times 10^{-7}$. It will be possible to increase this efficiency by optimizing the intensity ratio of incident $\omega$- and $2\omega$-pulses and tuning the spectral widths of odd- and even-parity excitons and their phase relaxation times by decreasing the temperature. In fact, it was reported that the maximum value of $\chi^{(3)}(-\omega; 0,0,\omega)$ dominating the electro-reflectance of this compound, increased with decreasing temperature mainly due to the sharpening of the band widths of exciton peaks[40,41,46]. Such an optimization of experimental conditions to enhance the terahertz radiation efficiency is a future subject.

## Discussion

Here, we discuss the physical mechanism of the observed terahertz radiation and the $\Delta t$ dependence of its phase, amplitude, and frequency in the third-order optical nonlinearity framework. The spectral widths of the excitation pulses were ignored for simplicity. At $\Delta t \sim 0$ fs, the third-order nonlinear polarization $P^{(3)}$ accounting for the terahertz radiation with a frequency $\omega_{THz}$ is expressed using two third-order nonlinear susceptibilities $\chi^{(3)}(-\omega_{THz}; \omega, \omega, -(2\omega - \omega_{THz}))$ and $\chi^{(3)}(-\omega_{THz}; -\omega, -\omega, 2\omega + \omega_{THz})$ as follows:

$$P^{(3)}(\omega_{THz}) \propto \chi^{(3)}(-\omega_{THz}; \omega, \omega, -2\omega + \omega_{THz})e^{-i2\omega\Delta t}E(\omega)E(\omega)E(-2\omega + \omega_{THz})$$

$$+ \chi^{(3)}(-\omega_{THz}; -\omega, -\omega, 2\omega + \omega_{THz})e^{i2\omega\Delta t}E(-\omega)E(-\omega)E(2\omega + \omega_{THz}).$$
(3)

The first term is related to the generation of a $(2\omega - \omega_{THz})$- pulse and a $\omega_{THz}$-pulse from two $\omega$- pulses, as illustrated in the left part of Fig. 4c. The second term is related to the generation of two $\omega$-pulses and an $\omega_{THz}$-pulse from a $(2\omega + \omega_{THz})$- pulse, as illustrated in the right part of Fig. 4c. As $\Delta t$ increases, the phase of the first and second terms change as $-2\omega\Delta t$ and $2\omega\Delta t$, respectively, so that the terahertz waves that arise from these two terms interfere, as shown in Fig. 4e. Since the magnitudes of $\chi^{(3)}(-\omega_{THz}; \omega, \omega, -(2\omega - \omega_{THz}))$ and $\chi^{(3)}(-\omega_{THz}; -\omega, -\omega, 2\omega + \omega_{THz})$ are approximately equal (Supplementary Note 5), $P^{(3)}(\omega_{THz})$ is a real number and is proportional to $\cos(2\omega\Delta t)$. As a result, the phase of the terahertz wave, $\phi$, should be 0 or $\pi$ as observed in Fig. 4a. The integrated Fourier amplitude should vary as $|\cos(2\omega\Delta t)|$ with a period $T = \frac{\pi}{2\omega}$, which can account for the $\Delta t$ dependence of $I_{THz}(\Delta t)$ at $\Delta t \sim 0$ fs in the resonant excitation (Fig. 5c) and for all the time region in the case of non-resonant excitation (Fig. 5f). Thus, the coherent responses without real exciton excitations are dominated by Eq. (3).

Next, we discuss the real exciton response for the resonant excitation. When the $\omega$-pulse is preceded by the $2\omega$-pulse ($\Delta t > 0$) as shown in Fig. 3c, even-parity excitons are generated first, and the first term of Eq. (3), illustrated in Fig. 4d, becomes dominant. Neglecting the second term of Eq. (3), $P^{(3)}(\omega_{THz}) \propto \exp(-2i\omega\Delta t)$ as shown in Fig. 4f. Therefore, the phase of the terahertz wave, $\phi$, should decrease proportionally to $\Delta t$, and $I_{THz}(\Delta t)$ does not exhibit any oscillations for $\Delta t > 50$ fs, as shown in Figs. 4b and 5c, respectively. More precisely, the $\omega$- and $2\omega$-pulses have a temporal width of approximately 100 fs. Therefore, they have a large overlap in the region up to $\Delta t \sim 50$ fs. This accounts for the oscillatory component that appears up to $\Delta t \sim 50$ fs as seen in Fig. 5c. Figure 5g shows the $\Delta t$ dependence of the spectral center of gravity of the Fourier power spectrum at $\Delta t$, which gives the maxima of $I_{THz}(\Delta t)$ in Fig. 5c. With increasing $\Delta t$, it gradually shifts from 1.25 THz to 1.55 THz, which implies that the origin of the terahertz

radiation changes from a coherent response at $t \sim 0$ fs to a real exciton response at $\Delta t > 50$ fs.

Such a high-frequency shift with increasing $\Delta t$ is characteristic of the terahertz radiation due to the third-order optical nonlinearity associated with well-defined even-parity and odd-parity excitonic states with different energies. Such a shift was not observed in the similar terahertz radiations in conventional 3D semiconductors, where excitonic effect is very weak. In the series of halogen-bridged nickel-chain compounds expressed as $[\text{Ni(chxn)}_2X]Y_2$, the energy difference between the even-parity and odd-parity excitons, $\hbar(\omega_e - \omega_o)$, can be widely changed from ~10 meV in $[\text{Ni(chxn)}_2\text{Br}]\text{Br}_2$ to ~90 meV by the choice of bridging halogen ion $X$ and counterion $Y$[40]. When the energies of the odd-parity and even-parity excitons in a 1D Mott insulator are far apart, e.g., $\hbar(\omega_e - \omega_o) \sim 90$ meV, it will be possible to generate a high-frequency terahertz pulse with the frequency of ~20 THz using the same technique. In that case, femtosecond laser pulses with short temporal widths having spectral widths larger than ~90 meV must be used as the excitation pulses, and the photon energies of $2\omega$- and $\omega$-pulses must be tuned to match the odd-parity exciton energy $\hbar\omega_o$ and half of the even-parity exciton energy, $\hbar\omega_e/2$, respectively. In addition, appropriate nonlinear optical crystals for the second harmonic generation and for the adjustment of time delay between $2\omega$- and $\omega$-pulses, $\Delta t$, are necessary. If these conditions are appropriately prepared, it may be possible to achieve the high-frequency terahertz pulse generation and to control the frequency over a wider range by varying $\Delta t$. This is an interesting issue in future.

Finally, we show that the present terahertz-radiation measurements can be performed to obtain information on the ultrafast dynamics of excitons. In this regard, we analyzed the time characteristic of the envelope of the oscillatory component that represented the coherent response, $a(\Delta t)$, and the envelope of the non-oscillatory component that reflected the real exciton response, $b(\Delta t)$. $a(\Delta t)$ is fundamentally a Gaussian profile $g(\Delta t)$ with a temporal width $t_w$, which is determined by the convolution of the intensity profile of the $\omega$-pulse and the amplitude profile of the $2\omega$-pulse. $b(\Delta t)$ can be expressed based on the exponential-decay function using the phase relaxation time of the even-parity exciton, $\tau_{\text{even}}$, convolved with $g(\Delta t)$. The expressions for $a(\Delta t)$ and $b(\Delta t)$ are reported in Supplementary Note 6. For the non-resonant excitation condition, $b(\Delta t) = 0$, and the envelope of $I_{\text{THz}}(\Delta t)$ is well reproduced by $a(\Delta t)$ for $t_w = 120$ fs, which is shown by the blue line in Figs. 5f and 5h. For the resonant exciton excitation condition, the envelope of $I_{\text{THz}}(\Delta t)$ and the envelope of the non-oscillatory component are well reproduced by the orange line showing $a(\Delta t) + b(\Delta t)$ and the green line showing $b(\Delta t)$, respectively, in Fig. 5c. $a(\Delta t)$ and $b(\Delta t)$ are also shown by the red and green lines, respectively, in Fig. 5h. $a(\Delta t)$ is characterized by $t_w = 60$ fs, which is shorter than $t_w = 120$ fs estimated in the non-resonant excitation condition. This suggests that the coherent response is suppressed when the real exciton excitations increase within the temporal width of the laser pulses. The non-oscillatory component $b(\Delta t)$ is characterized by the phase relaxation time of the even-parity exciton, $\tau_{\text{even}} = 38$ fs, which is sufficiently short compared to its relaxation time to the ground state of approximately 1 ps[60]. This means that the effect of the relaxation to the ground state of the even-parity exciton on the time characteristic of $b(\Delta t)$ can be neglected. Furthermore, $\tau_{\text{even}}$ is much shorter than the exciton phase relaxation times in inorganic semiconductors, which are typically 10 ps[61]. The observed short phase relaxation time would be characteristic of strongly correlated electron systems.

In this study, we demonstrated the highly effective generation of terahertz waves from a typical 1D Mott insulator $[\text{Ni(chxn)}_2\text{Br}]\text{Br}_2$ by exciting odd-parity and even-parity excitons using two-color femtosecond pulses. It was shown that the phase,

amplitude, and frequency of the terahertz waves can be controlled by the fine tuning of the generation time difference of two excitons. We also demonstrated that this controllability of the terahertz waves is based on the quantum interference of two third-order nonlinear optical responses associated with terahertz radiation, which are characterized by $\chi^{(3)}(-\omega_{\text{THz}}; \omega, \omega, -(2\omega - \omega_{\text{THz}}))$ and $\chi^{(3)}(-\omega_{\text{THz}}; -\omega, -\omega, 2\omega + \omega_{\text{THz}})$. When the delay of the generation time of the odd-parity excitons relative to the even-parity excitons is small, the two nonlinear processes interfere, and the radiation intensity oscillates depending on the delay. By increasing the delay time, the nonlinear process characterized by $\chi^{(3)}(-\omega_{\text{THz}}; \omega, \omega, -(2\omega - \omega_{\text{THz}}))$ becomes dominant, resulting in non-oscillating terahertz radiation. By analyzing its time characteristic, we evaluated the phase relaxation time of the excitons to be approximately 40 fs, which is much shorter than the typical value (10 ps) in inorganic semiconductors. This approach can be applied to various semiconductors to obtain the phase information of excitonic states if a wider frequency range of light pulses and an appropriate detection technique are used.

## Online content

## Methods

### Sample preparation

Single crystals of $[\text{Ni(chxn)}_2\text{Br}]\text{Br}_2$ were grown using an electrochemical method[47]. The obtained crystals were platelets with a large $bc$ plane of approximately 1 mm × 1 mm and a thickness of approximately 0.5 mm.

### Steady-state optical reflection spectroscopy measurements

The polarized reflectivity ($R$) spectrum of a single crystal of $[\text{Ni(chxn)}_2\text{Br}]\text{Br}_2$ was measured on the $bc$ plane using a specially designed spectrometer with a grating monochromator that operated in the visible and near-infrared region from 0.5 to 5.0 eV, and a Fourier-transform infrared spectrometer in the mid-infrared region from 0.08 to 1.2 eV. The spectra of the real ($\varepsilon_1$) and imaginary ($\varepsilon_2$) parts of the complex dielectric constant were calculated based on the polarized $R$ spectra using the Kramers–Kronig transformation.

### Terahertz radiation measurements

Terahertz radiation measurements were performed using a Ti: sapphire regenerative amplifier (RA) as the light source, with a central photon energy of 1.55 eV, repetition rate of 1 kHz, pulse duration of 100 fs, and pulse energy of 2 mJ. The schematic of the optical system is shown in Supplementary Fig. 1a. The output of the RA was divided into two beams. One of the outputs was input to an optical parametric amplifier to obtain the $\omega$-pulse ($\hbar\omega = 0.64$ eV or 0.585 eV). The $2\omega$-pulse was generated by introducing the $\omega$-pulse to a $\beta$-$\text{BaB}_2\text{O}_4$ ($\beta$-BBO) crystal. Terahertz waves were generated by focusing both the $\omega$-pulse and the $2\omega$-pulse, which are polarized perpendicular to each other, onto the sample simultaneously. Delay time $\Delta t$ of the $\omega$-pulse relative to the $2\omega$-pulse is controlled by passing two pulses through an $\alpha$-BBO crystal with a thickness of 1 mm[62]; the optical path lengths of two pulses can be changed by rotating the $\alpha$-BBO crystal. Electric-field waveforms of terahertz pulses generated in the reflection configuration were measured using an electro-optic sampling (EOS) method with a 1-mm thick ZnTe crystal[63]. The other RA output was used as a sampling pulse for the EOS. The details of the

terahertz radiation measurements and method for controlling the delay time $\Delta t$ are reported in Supplementary Notes 1 and 2, respectively.

The electric-field amplitudes of terahertz radiations were evaluated from the electro-optic signal $\delta$ (ref. [64]) by using the following equation[3,12]:

$$E_{THz} = \frac{\lambda_0}{2\pi l n_0^3 r_{41} t_{ZnTe}} \sin^{-1}(\delta) \qquad (4)$$

where $l = 1\,mm$, $n_0 = 2.87$ (ref. [64]), $r_{41} = 4.04\,pm/V$ (ref. [65]), and $t_{ZnTe} = 2/(n_{THz} + 1) = 0.48$ ($n_{THz} = 3.17$: ref. [66]), are the thickness, the refractive index of the sampling pulse with the wavelength $\lambda_0 = 800\,nm$, the electro-optical coefficient at the same wavelength, and the Fresnel transmission coefficients at 1 THz of ZnTe, respectively. The intensity of terahertz radiation, $U$, was evaluated from $E_{THz}(t)$ by using the following formula:

$$U = \frac{2S}{Z_0} \int_{-\infty}^{\infty} E_{THz}^2(t)dt, \qquad (5)$$

where $Z_0 = 377\,\Omega$ is the vacuum impedance and $S = 0.95\,mm^2$ is the area with half the total intensity of terahertz radiation.

## Data availability
The raw data generated in this study are provided in the Source Data file. Source data are provided with this paper.

## Code availability
The codes are available from the corresponding author upon reasonable request.

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

## Acknowledgements

We thank Prof. T. Tohyama and K. Shinjo for enlightening discussions. This work was partly supported by a Grant-in-Aid for Scientific Research from the Japan Society for the Promotion of Science (JSPS) (Project Numbers: JP20K03801, JP21H04988) and CREST (JPMJCR1661), Japan Science and Technology Agency. Ta.Mo. was supported by JSPS through Program for Leading Graduate Schools (MERIT) and JSPS Research Fellowships for Young Scientists.

## Author contributions

Ta.Mi., A.K., T.I., Ta.Mo. and S.Y. built the terahertz-radiation measurement system. A.K., T.I. and Ta.Mi. carried out the terahertz-radiation measurements of the bromine-bridged Ni-chain compound. H.O. prepared the sample. A.K., T.I. and Ta.Mi. analyzed the experimental data. Ta.Mi. and H.O. coordinated the study. The manuscript was written by Ta.Mi. and H.O. with inputs from all authors.

## Competing interests

The authors declare no competing interests.
