## [Peer Review File · Nature Communications]

REVIEWER COMMENTS

Reviewer #1 (Remarks to the Author):

Miyamoto et al present a study about THz generation by quantum interference of odd- and even-parity excitons in one dimensional Mott insulator made of Br-bridged Ni-chain compound. The population of the two nearly-degenerate excitonic states, exhibiting a small energy difference corresponding to THz photons, is resonantly enabled by two-color pulsed laser excitation. Two-photon absorption at frequency ω excites the even-parity exciton, and single-photon absorption at doubled-frequency 2ω excites the one of odd-parity. Thus, the THz generation process in this case demonstrates a third-order nonlinear light-matter interaction. Importantly, the phase, amplitude and central frequency of the generated THz radiation can all be controlled by controlling the time-delay between the two excitation laser pulses. The experimental results are modeled and well-justified by a convincing quantum theory for the quantum interference between the two excitons. The concept was further validated by additional measurements with non-resonant laser excitations at lower photon energies, which showed distinguished dependence on the time delay between the two driving laser pulses.

The results are highly interesting, well-presented, and clearly and satisfactorily interpreted. The study is, indeed, of great relevance for THz science and technology. The data quality is high, and the manuscript is well-structured and well-written in good English. The list of references is satisfying with prior and related works adequately referenced.

I just wish that the authors comment on the optical-to-THz power conversion efficiency, and how this efficiency can be optimized. Additionally, the colors of the curve in Fig. 5h for the case of non-resonant excitation and its corresponding legend-labeling should match. With these minor additions to the manuscript, I will be glad to see this work published in Nature Communications soon.

Reviewer #2 (Remarks to the Author):

Miyamoto et al. report THz emission in a 1D Mott insulator by using two-color laser pulses. The purpose of research is to control the phase and frequency of monocyclic terahertz pulse by quantum interference of odd- and even-parity excitons. By controlling a time delay between the two pulses, the authors demonstrated that the CEP control of THz pulse and reported the peak shift of the

Fourier amplitude spectra. The authors also discuss the mechanism in terms of a coherent response and a real exciton response.

The experimental results look solid, and the interpretations are grounded. However, I feel that the novelty and importance of this work is not suited for publication in Nature Communications for the following reasons.

(i) The method employed in this work looks identical to the well-known technique named two-color quantum interference control (QUIC). Arbitrary control of the CEP of THz emission by adjusting the two pulses has been already reported in conventional semiconductors. See e.g., Phys. Rev. 77, 085201 (2008). Not only the phase, the vectorial control of the current and the spin current is also possible in this technique. See e.g., Phys. Rev. Lett. 96, 246601 (2006) and Nature Photon. 14, 680 (2020). Although there are many other related literatures for this technique, they are not cited in this work.

As far as I understand, the use of even- and odd-parity excitons in this work is not essential for this technique. Because the two exciton levels are close and the spectra are broadened, this material just shows one- and two-photon absorptions at the similar photon energies. Therefore, the same response should occur for the interband transition in conventional semiconductors where a not-well-defined parity allows both absorptions.

(ii) The authors state that the phase and frequency control “are challenging to obtain in the optical rectification (OR) framework since the OR process is a simple classical electromagnetic phenomenon.”

I do not agree with this. All the nonlinear responses including OR is the quantum mechanical phenomenon in the microscopic point of view, and one can just also recognize OR in a classical phenomenological picture. THz emission by OR is a kind of differential frequency generation (DFG) process. DFG using two different pulses can also tune both the center frequency and the CEP of THz emission by adjusting the time delay between the two pulses. See e.g., Opt. Lett. 33, 2767 (2008), Nature Photon. 8, 119 (2014), Opt. Lett. 42, 129 (2017).

(iii) The main text begins with the interest in control of solids by using intense THz pulse. In the abstract, the authors state that “an intense terahertz wave is generated.” Nevertheless, I could not find any value of the THz field strength achieved in this work. In general, the third-order nonlinearity is weaker than the conventional second-order response, and therefore how strong field is available can be the disadvantage for this work. The authors should show the peak value of the THz field.

(iv) In the abstract, the authors also state that “one can control all of the phase, frequency, and amplitude of terahertz waves by adjusting the creation-time difference of two excitons”. Figure 5b shows that the peak of the Fourier-amplitude spectra shifts from 1.2 to 1.5 THz. However, the data

are normalized, and the bumpy spectral shape for the weakened condition looks like that the THz output is just suppressed due to a deconstructive interference. "Control of frequency" sounds overstatement.

Summary of changes

We have thoroughly revised both the manuscript (MS) and Supplementary Information (SI).

- (1) Taking the format in *Nature Communications* into account, Extended Data Figure 1 in the previous MS have been moved to Supplementary Note 4 in the revised SI.
- (2) Taking the format in *Nature Communications* into account, Extended Data Figure 2 and the related sentences in the Methods section in the previous MS have been moved to Supplementary Note 1 in the revised SI.
- (3) Taking comment (1) of the Reviewer #1 and comments (1)-(3) of the Reviewer #2 into account, we have added a new subsection “**Comparison of terahertz radiations from [Ni(chxn)₂Br]Br₂ and conventional semiconductors**” in the Results section. We have also added the related figure, Fig. 6 in the revised MS. In addition, we have added the evaluation method of the electric-field amplitudes and the intensity of terahertz radiations in the Methods section.
- (4) Taking comment (2) of the Reviewer #1 into account, the order of the legend labels in Fig. 5h has been changed.
- (5) Taking comment (1) of the Reviewer #2 into account, we have revised the latter part of the introduction to clarify the novelty and aim of our study in condensed matter physics and optical science.
- (6) Taking comment (1) of the Reviewer #2 into account, we have added the sentences to emphasize the uniqueness of the controllability of electric-field amplitude and phase demonstrated in our study.
- (7) Taking comment (2) of the Reviewer #2 into account, we have deleted some misleading explanations about the optical rectification process and some descriptions about the other methods to control the phases of terahertz waves.
- (8) Taking comment (4) of the Reviewer #2 into account, we have added the explanations about the control of the terahertz radiation frequency.
- (9) We have also added the related references and renumbered some references in the revised MS and SI.

All the changed parts are marked in yellow in Marked MS.pdf and Marked SI.pdf.

Replies to comments of Reviewer #1

We would like to thank Reviewer #1 for his/her careful reading of our paper and valuable comments. We would like to reply to his/her comments below.

Comment (1) of Reviewer #1

I just wish that the authors comment on the optical-to-THz power conversion efficiency, and how this efficiency can be optimized.

Reply to comment (1) of Reviewer #1

We would like to thank Reviewer #1 for his/her important comment. Reviewer #2 also questioned about the electric-field amplitude of the detected terahertz radiation. In addition, Reviewer #2 advised us to refer to previous related studies on conventional semiconductors. Therefore, we have performed the additional experiments to evaluate the conversion efficiency of the terahertz radiation from two near-infrared pulses and the absolute value of the electric-field amplitude of the terahertz radiation. Furthermore, we have compared the result with those in conventional semiconductors. These results and discussions are detailed in a new subsection “**Comparison of terahertz radiations from [Ni(chxn)₂Br]Br₂ and conventional semiconductors**” in the Results section, which is shown below.

[Line 14 of page 9 to line 4 of page 14]

“Comparison of terahertz radiations from [Ni(chxn)₂Br]Br₂ and conventional semiconductors

As mentioned in the introductory part, in conventional semiconductors with inversion symmetry such as Si, Ge, InP, and GaAs, terahertz radiations based upon the third-order nonlinear optical process expressed by $P(0) = \frac{3}{4}\epsilon_0\chi^{(3)}(0; \omega, \omega, -2\omega)E(\omega)E(\omega)E(-2\omega)$ using $\omega -$ and $2\omega -$ pulses have also been reported. In those semiconductors, the two-photon excitation by the $\omega -$ pulse and one-photon excitation by the $2\omega -$ pulse are both due to the interband transitions. It is interesting to compare the efficiency of the terahertz radiation from [Ni(chxn)₂Br]Br₂ with those from those semiconductors, which originate from the similar third-order nonlinear optical processes. For this subject, we have measured the terahertz radiations from [Ni(chxn)₂Br]Br₂, Si, and Ge with two incident pulses of $\hbar\omega = 0.64$ eV and $2\hbar\omega = 1.28$ eV in the same experimental setup with the reflection configuration (see Supplementary Note 1). For Si and Ge, crystals cut out of the (100) plane were used. The

excitation fluences of the ω – and 2ω – pulses are 1.77 mJ/cm^2 and 0.11 mJ/cm^2 , respectively. The ω – pulse was used for the two-photon excitation, and its fluence was much higher than that of the 2ω – pulse used for the one-photon excitation.

In Fig. 6, the electric field waveforms of the terahertz radiations from Si and Ge are shown by the green and red lines, respectively, together with that emitted from $[\text{Ni}(\text{chxn})_2\text{Br}]\text{Br}_2$ (the black line). The electric field amplitude (intensity) of the terahertz radiation from $[\text{Ni}(\text{chxn})_2\text{Br}]\text{Br}_2$ is approximately 2.4 (5.8) times higher than that from Ge and 50 (2500) times higher than that from Si. The terahertz radiation from Si is much smaller than that from Ge. It is because the direct interband transition cannot be excited by the 1.28 eV pulse in Si but can be excited in Ge.

Next, we compare the terahertz radiation efficiency of $[\text{Ni}(\text{chxn})_2\text{Br}]\text{Br}_2$ and Ge more precisely. In our experiments, the polarizations of ω – and 2ω – pulses are orthogonal (see Supplementary Fig. 1b). In the experiments on $[\text{Ni}(\text{chxn})_2\text{Br}]\text{Br}_2$, they are tilted by ± 45 degrees with respect to the 1D Ni chain (see Supplementary Note 1). If both polarizations are parallel to the 1D Ni chain (the b -axis) while maintaining the intensities of two pulses, the electric field amplitude of the terahertz radiation is expected to increase by a factor of $2\sqrt{2}$ and its intensity by a factor of 8. On the other hand, in Ge, the terahertz radiation intensity for the polarizations of two excitation pulses parallel to each other is roughly 1.1 times higher than that for their polarizations perpendicular to each other³³. Therefore, the intensity of the terahertz radiation from $[\text{Ni}(\text{chxn})_2\text{Br}]\text{Br}_2$ is estimated to be approximately 42 times ($= 5.8 \times 8 / 1.1$) as large as that from Ge.

More strictly, it is necessary to discuss the terahertz radiation efficiency by taking the penetration depths of the incident pulses into account. Since the penetration depth of the 2ω – pulse, $l(2\omega)$, is much shorter than that of the ω – pulse, $l(\omega)$, the terahertz radiation is emitted only from a region as deep as $l(2\omega)$. Therefore, it is reasonable to consider that an electric-field amplitude of the terahertz radiation is proportional to $l(2\omega)$ and its intensity is proportional to $[l(2\omega)]^2$. $l(2\omega)$ is approximately 20 nm in $[\text{Ni}(\text{chxn})_2\text{Br}]\text{Br}_2$ and approximately 500 nm in Ge⁵⁰. To compare the terahertz radiation efficiency per unit length between $[\text{Ni}(\text{chxn})_2\text{Br}]\text{Br}_2$ and Ge, it is necessary to multiply the terahertz radiation intensity in the former approximately 600 times. Taking this factor into account, the terahertz radiation efficiency per unit length in $[\text{Ni}(\text{chxn})_2\text{Br}]\text{Br}_2$ is estimated to be 2.6×10^4 times as large as that of Ge. This extremely high efficiency of the terahertz radiation in $[\text{Ni}(\text{chxn})_2\text{Br}]\text{Br}_2$ is characteristic of 1D Mott insulators, in which third-order nonlinear optical responses associated with nearly degenerate odd- and even-parity excitons are very large. Such large third-order nonlinear optical responses originate from the effects of strong electron correlation and one-dimensionality^{36–38}.

In CdSe and CdTe, it has been attempted to generate the phase-controlled terahertz pulses by the excitations of $1s$ and $2p$ excitons with different symmetries³⁴. However, the exciton binding energies of these compound semiconductors are still small; the binding energy of the lowest exciton is approximately 15 meV in CdSe and 11 meV in CdTe. In those compounds, therefore, it is difficult to resonantly excite only those excitons and the generated terahertz waves necessarily include two types of the components, which originate from the interband transitions and from the exciton transitions. As a result, when varying the phase difference between the ω – and 2ω – pulses, both amplitude and phase of terahertz radiation are simultaneously changed, and the continuous control of the electric-field phase of the terahertz radiation with the electric-field amplitude fixed, as demonstrated in $[\text{Ni}(\text{chxn})_2\text{Br}]\text{Br}_2$, had not been achieved^{32,34}. In addition, these compounds are three-dimensional (3D) semiconductors and the oscillator strength of the lowest exciton is not so large. Therefore, the intensity of the terahertz radiation caused by the exciton transitions is much smaller than that caused by the interband transitions. In our study, in $[\text{Ni}(\text{chxn})_2\text{Br}]\text{Br}_2$, the terahertz radiation occurs only via the quantum interference between the odd-parity and even-parity excitons and is not affected by the interband transitions. It is because the binding energies of the odd-parity exciton and even-parity exciton, which are approximately 90 meV and 80 meV, respectively⁴⁰, are sufficiently large as compared to the spectral width of the excitation pulses (approximately 20 meV) and the oscillation strength is concentrated on the lowest odd-parity exciton reflecting the 1D nature of the electronic system.

Using the same experimental setup, we also measured the terahertz radiation from ZnTe, which is a second-order nonlinear optical crystal without inversion symmetry and widely used to generate terahertz electromagnetic waves via the OR process described by $P(0) = 2\varepsilon_0\chi^{(3)}(0; \omega, -\omega)E(\omega)E(-\omega)$ ^{51,52}. In general, the third-order nonlinear optical effect described by $P(0) = \frac{3}{4}\varepsilon_0\chi^{(3)}(0; \omega, \omega, -2\omega)E(\omega)E(\omega)E(-2\omega)$ should be much smaller than the second-order nonlinear optical effect. Since the excitation fluence of the ω – pulse is approximately 16 times higher than that of the 2ω – pulse, it can be considered that the terahertz radiation occurs via the OR induced by the ω – pulse in ZnTe. The electric field waveform of the terahertz radiation from ZnTe with the (110) plane cut out is shown by the blue line in Fig. 6. The electric field amplitude of the terahertz radiation emitted from $[\text{Ni}(\text{chxn})_2\text{Br}]\text{Br}_2$ is 5.4 times as large as that from ZnTe. Although the difference is that the former used resonant exciton excitations and the latter used non-resonant excitations in the transparent region, this result also indicates that the efficiency of the terahertz radiation from $[\text{Ni}(\text{chxn})_2\text{Br}]\text{Br}_2$ is very high.

Controlling the phase and frequency of a terahertz pulse might be possible using a second-order nonlinear optical effect. In fact, it was reported that a mid-infrared pulse can be generated from two near-infrared pulses with different frequencies via the differential frequency generation (DFG) process^{53–55}. In those studies, the tuning of phase and frequency of the mid-infrared pulse was achieved by adjusting the time difference between two near-infrared pulses. However, this technique would be somewhat difficult to apply for a terahertz radiation. It is because a terahertz radiation due to the simple single-pulse OR process occurs by each incident pulse, making it difficult to control the phase and frequency of the terahertz radiation via the DFG process. In addition, in the DFG method, a complicated measurement system was necessary to prepare two optical paths with precisely adjusted lengths for two different near-infrared pulses. In contrast, the method using the third-order optical nonlinearity reported here requires only two pulses with frequencies of ω and 2ω . The latter pulse can be easily obtained from the former pulse using the SHG process within the same optical path (see Supplementary Fig. 1a), so the measurement system is much more simplified.

Finally, we estimate the energy conversion efficiency from two excitation pulses to a terahertz pulse in $[\text{Ni}(\text{chxn})_2\text{Br}]\text{Br}_2$ with our method using the third-order optical nonlinearity. In our experiments, the total energy of the ω – and 2ω –pulses incident to the sample is 1.8×10^{-6} J, while the energy of the terahertz radiation emitted from $[\text{Ni}(\text{chxn})_2\text{Br}]\text{Br}_2$ is 1.3×10^{-13} J (see Methods section). Considering the correction factor of 8 mentioned above, we obtained the energy efficiency of 5.8×10^{-7} . It will be possible to increase this efficiency by optimizing the intensity ratio of incident ω – and 2ω –pulses and tuning the spectral widths of odd- and even-parity excitons and their phase relaxation times by decreasing the temperature. In fact, it was reported that the maximum value of $\chi^{(3)}(-\omega; 0, 0, \omega)$ dominating the electro-reflectance of this compound, increased with decreasing temperature mainly due to the sharpening of the band widths of exciton peaks^{38,40,43}. Such an optimization of experimental conditions to enhance the terahertz radiation efficiency is a future subject.”

Related to those revisions, we have added several references listed below.

32. Costa, L., Betz, M., Spasenović, M., Bristow, A. D. & Van Driel, H. M. All-optical injection of ballistic electrical currents in unbiased silicon. *Nat. Phys.* **3**, 632–635 (2007).
33. Spasenović, M., Betz, M., Costa, L. & Van Driel, H. M. All-optical coherent control of electrical currents in centrosymmetric semiconductors. *Phys. Rev. B* **77**, 085201 (2008).

34. Sames, C., Ménard, J. M., Betz, M., Smirl, A. L. & Van Driel, H. M. All-optical coherently controlled terahertz ac charge currents from excitons in semiconductors. *Phys. Rev. B* **79**, 045208 (2009).
50. Dash, W. C. & Newman, R. Intrinsic optical absorption in single-crystal germanium and silicon at 77 K and 300 K. *Phys. Rev.* **99**, 1151–1155 (1955).
51. Nahata, A., Weling, A. S. & Heinz, T. F. A wideband coherent terahertz spectroscopy system using optical rectification and electro-optic sampling. *Appl. Phys. Lett.* **69**, 2321–2323 (1996).
52. Blanchard, F. et al. Generation of 1.5 μJ single-cycle terahertz pulses by optical rectification from a large aperture ZnTe crystal. *Opt. Express* **15**, 13212–13220. (2007).
53. Sell, A., Leitenstorfer, A. & Huber, R. Phase-locked generation and field-resolved detection of widely tunable terahertz pulses with amplitudes exceeding 100 MV/cm. *Opt. Lett.* **33**, 2767–2769 (2008).
54. Schubert, O. et al. Sub-cycle control of terahertz high-harmonic generation by dynamical Bloch oscillations. *Nat. Photon.* **8**, 119–123 (2014).
55. Liu, B. et al. Generation of narrowband, high-intensity, carrier-envelope phase-stable pulses tunable between 4 and 18 THz. *Opt. Lett.* **42**, 129–131 (2017).

In addition, we have added the evaluation method of the electric-field amplitude and the intensity of terahertz radiations in the Methods section.

[Line 7 to 18 of page 25]

“The electric-field amplitudes of terahertz radiations were evaluated from the electro-optic signal δ (ref. 58) by using the following equation^{3,12}:

$$E_{\text{THz}} = \frac{\lambda_0}{2\pi l n_0^3 r_{41} t_{\text{ZnTe}}} \sin^{-1}(\delta)$$

where $l = 1 \text{ mm}$, $n_0 = 2.87$ (ref. 59), $r_{41} = 4.04 \text{ pm/V}$ (ref. 60), and $t_{\text{ZnTe}} = 2/(n_{\text{THz}} + 1) = 0.48$ ($n_{\text{THz}} = 3.17$: ref. 61), are the thickness, the refractive index of the sampling pulse with the wavelength $\lambda_0 = 800 \text{ nm}$, the electro-optical coefficient at the same wavelength, and the Fresnel transmission coefficients at 1 THz of ZnTe, respectively. The intensity of terahertz radiation, U , was evaluated from $E_{\text{THz}}(t)$ by using the following formula:

$$U = \frac{2S}{Z_0} \int_{-\infty}^{\infty} E_{\text{THz}}^2(t) dt,$$

where $Z_0 = 377 \Omega$ is the vacuum impedance and $S = 0.95 \text{ mm}^2$ is the area with half

the total intensity of terahertz radiation.”

We have also added the related references in the revised MS.

59. Sato, K. & Adachi, S. Optical properties of ZnTe. *J. Appl. Phys.* **73**, 926–931 (1993).
60. Wu, Q. & Zhang, X. –C. Ultrafast electro-optic field sensors. *Appl. Phys. Lett.* **68**, 1604–1606 (1996).
61. Schall, M., Walther, M. & Jepsen, P. U. Fundamental and second-order phonon processes in CdTe and ZnTe. *Phys. Rev. B* **64**, 094301 (2001).

Comment (2) of Reviewer #1

Additionally, the colors of the curve in Fig. 5h for the case of non-resonant excitation and its corresponding legend-labeling should match.

Reply to comment (2) of Reviewer #1

The colors of the legend labels in Fig. 5h correspond correctly to the colors of the lines. To avoid confusing the readers, we have changed the order of the legend labels in the order of pink, blue, and green to match the positions of three curves.

Replies to comments of Reviewer #2

We would like to thank Reviewer #2 for his/her careful reading of our paper and thoughtful comments. We divide his/her comments to five parts (0)-(4) and would like to reply to each of them below.

Comment (0) of Reviewer #2

Miyamoto et al. report THz emission in a 1D Mott insulator by using two-color laser pulses. The purpose of research is to control the phase and frequency of monocyclic terahertz pulse by quantum interference of odd- and even-parity excitons. By controlling a time delay between the two pulses, the authors demonstrated that the CEP control of THz pulse and reported the peak shift of the Fourier amplitude spectra. The authors also discuss the mechanism in terms of a coherent response and a real exciton response.

The experimental results look solid, and the interpretations are grounded. However, I feel that the novelty and importance of this work is not suited for publication in Nature Communications for the following reasons.

Reply to comment (0) of Reviewer #2

The referee has doubts about the novelty and importance of our study. We would like to answer this comment in detail in the reply to comment (1).

Comment (1) of Reviewer #2

The method employed in this work looks identical to the well-known technique named two-color quantum interference control (QUIC). Arbitrary control of the CEP of THz emission by adjusting the two pulses has been already reported in conventional semiconductors. See e.g., Phys. Rev. 77, 085201 (2008). Not only the phase, the vectorial control of the current and the spin current is also possible in this technique. See e.g., Phys. Rev. Lett. 96, 246601 (2006) and Nature Photon. 14, 680 (2020). Although there are many other related literatures for this technique, they are not cited in this work.

As far as I understand, the use of even- and odd-parity excitons in this work is not essential for this technique. Because the two exciton levels are close and the spectra are broadened, this material just shows one- and two-photon absorptions at the similar photon energies. Therefore, the same response should occur for the interband transition in conventional semiconductors where a not-well-defined parity allows both absorptions.

Reply to comment (1) of Reviewer #2

We would like to thank Reviewer #2 for his/her important comment. In our manuscript

(MS) initially submitted, due to our lack of the research of related studies, we did not cite the previous studies on the current generations by the optical rectification using third-order nonlinear optical effects in conventional semiconductors and the resulting terahertz radiations. As pointed out by Reviewer #2, we should appropriately cite those studies at the beginning of our manuscript. We also should make clear the difference between our present study on a 1D Mott insulator and the previous studies on conventional semiconductors and highlight the new findings and achievements in our study. Therefore, we have substantially revised our MS. Let us explain those revisions below.

In our study of the terahertz radiation in 1D Mott insulators, we used the third-order nonlinear optical response expressed by $P(0) = \frac{3}{4}\epsilon_0\chi^{(3)}(0; \omega, \omega, -2\omega)E(\omega)E(\omega)E(-2\omega)$, with two incident pulses with the frequencies ω and 2ω . The framework of this nonlinear optical response is the same as that used previously in the conventional semiconductors as Reviewer #2 pointed out. However, there are three important differences between the terahertz radiation of 1D Mott insulators and those of conventional semiconductors, as follows.

The first difference is that in the 1D Mott insulator under study, two excitonic states with even- and odd-parity with slightly different energies are generated by the ω –pulse via the two-photon absorption process and by the 2ω –pulse via the one-photon absorption process, respectively. The even-parity exciton has the higher energy than the odd-parity exciton. In those well-defined excitonic states, the interference of the polarizations of two excitons can be understood by a clear physical picture, that is, the oscillation of electrons shown by the thick green curve in Fig. 1d. In this situation, sophisticated controls of terahertz radiations become possible. As shown in Fig. 3(b,d) and Fig. 4(a,b), the phase and amplitude of the terahertz radiation change depending on the time difference Δt between the incident ω – and 2ω –pulses. When Δt is small, the two processes presented in Fig. 4c occur simultaneously and interfere with each other as shown in Fig. 4e. In this case, the amplitude of the terahertz radiation is continuously changed with Δt , while its phase is fixed at $+\pi$ or $-\pi$. On the other hand, when Δt increases up to ~ 60 fs, only one of the two processes is effective as shown in Fig. 4d. In this case, the electric-field amplitude of the terahertz radiation remains constant as seen in Fig. 3d and only its phase is changed continuously with Δt as shown in Fig. 3d and Fig. 4b. Namely, by setting the appropriate value of the interval between the incident ω – and 2ω –pulses, that is, the creation time difference of two excitons with the accuracy

of ten attoseconds, we can select the continuous control of the amplitude of the terahertz radiation with the phase fixed to $\pm\pi$ or the continuous control of the phase of the terahertz radiations from $-\pi$ to $+\pi$ with the amplitude fixed.

In the terahertz radiation phenomena previously studied in the conventional semiconductors, photo-excitations fundamentally generate interband transitions. Therefore, the dynamics of photoexcited carriers under the influence of their complicated scattering processes with other carriers and phonons would affect the temporal behaviors of induced currents. This makes it difficult to clarify the detailed terahertz radiation mechanism, and possibly suppresses terahertz radiation efficiencies. In II-VI semiconductors of CdSe and CdTe, the possibility of controlling the phase of terahertz radiation has been attempted by changing the temporal difference Δt between the excitations of $1s$ and $2p$ excitons [C. Sames et al. *Phys. Rev. B* **79**, 045208 (2009)]. However, the exciton binding energies of these compound semiconductors are still small; the binding energy of the lowest exciton is approximately 15 meV in CdSe and 11 meV in CdTe. In those compounds, therefore, it is difficult to resonantly excite only the excitonic states and the generated terahertz waves necessarily include two types of the components, which originate from the interband transitions and from the exciton transitions. As a result, when varying the phase difference between the ω – and 2ω – pulses, both amplitude and phase of terahertz radiation are simultaneously changed, and the continuous control of the electric-field phase of the terahertz radiation with the electric-field amplitude fixed, as demonstrated in $[\text{Ni}(\text{chxn})_2\text{Br}]\text{Br}_2$, had not been achieved.

We would like to emphasize that the 1D Mott insulator having the well-defined excitonic states gives an idealistic platform to investigate the terahertz radiation via the third-order optical rectification. In this paper, using this platform of 1D Mott insulators, we demonstrated the characteristic features of the terahertz radiation via the third-order optical nonlinearity associated with two nearly degenerate excitonic states with different parities. Furthermore, in 1D Mott insulators, the central frequency of the terahertz radiation can be controlled as shown in Fig. 5(b,g). The mechanism of this frequency control is discussed in detail in response to comment (2) of Reviewer #2.

The second difference between the terahertz radiation from the 1D Mott insulator and those from conventional semiconductors is that the efficiency of the terahertz radiation from the former is orders of magnitude higher than that from the latter. This point is discussed in detail in the response to comment (3) of Reviewer #2.

The third difference between the terahertz radiation from the 1D Mott insulator and those from conventional semiconductors is that the phase relaxation time of excitonic states can be determined in the former. It is an important subject to clarify the phase relaxation time of an exciton in Mott insulators. It is because the exciton in Mott insulators is a bound state of doublon and holon and is essentially different from excitons in band insulators or conventional semiconductors. To our best of knowledge, however, there has been no experimental evaluation of the phase relaxation time of an exciton in Mott insulators.

To summarize, using a 1D Mott insulator of the bromine-bridged nickel-chain compound, our paper reports three important results, which have not been reported so far; first, the demonstration of the characteristic phase and amplitude controllability of the terahertz radiation by changing the temporal difference of two excitation pulses; second, the demonstration of the high generation efficiency of the terahertz radiation originating from the large third-order optical nonlinearity characteristic of 1D Mott insulators; third, the evaluation of the phase relaxation time of excitons in 1D Mott insulators. We believe that our study presents novel and important phenomena in condensed matter physics and optical science.

To refer to the previous studies on the terahertz radiations in conventional semiconductors related to our study and make clear the aim of our present study on a 1D Mott insulator, we revised the introductory parts as follows.

[Line 6 of page 2 to line 6 of page 3]

“Terahertz pulses with variable phase, frequency and amplitude are useful for controlling the electronic states of matter^{25–28} but are challenging to achieve in the simple optical rectification (OR) framework in which a second-order nonlinear optical crystal is excited with a femtosecond laser pulse in the transparent region. An effective method for this is to use an OR process based upon the third-order optical nonlinearity associated with two pulses with the frequencies of ω and 2ω , which is expressed by $P(0) = \frac{3}{4}\epsilon_0\chi^{(3)}(0; \omega, \omega, -2\omega)E(\omega)E(\omega)E(-2\omega)$. Here, $E(\omega)$ and $E(-2\omega)$ are the electric fields of ω – and 2ω – pulses, respectively. This process causes a finite photocurrent even in centrosymmetric materials^{29–31} and therefore may extend the possibility to generate terahertz electromagnetic waves effectively. In fact, in conventional semiconductors, i.e., IV semiconductors of silicon (Si) and germanium (Ge), III-V semiconductors of GaAs and InP, and II-VI semiconductors of CdSe and CdTe, terahertz

radiations due to this mechanism were detected^{32–34}. In these phenomena, photo-excitations fundamentally generate interband transitions so that the dynamics of photoexcited carriers under the influence of their complicated scattering processes with other carriers and phonons would affect the temporal behaviors of induced currents. This makes it difficult to clarify the detailed terahertz radiation mechanism, and possibly suppresses terahertz radiation efficiencies. In II-VI semiconductors, the possibility of controlling the phase of terahertz radiation using a transition between different excitonic states has also been investigated³⁴. However, due to the weak excitonic effect, the terahertz radiation associated with the excitonic transitions is much smaller than that originating from the interband transitions. We expect that more advanced controls of terahertz radiations in the framework of the third-order optical nonlinearity, namely the generations of phase-, frequency-, and amplitude-adjustable terahertz pulses with high efficiency would be possible by utilizing specific materials having well-defined wavefunctions of photoexcited states and large values of related $\chi^{(3)}(0; \omega, \omega, -2\omega)$.”

We have quoted the following papers on the studies of photocurrents by two-color quantum interference control in conventional semiconductors.

29. Haché, A. et al. Observation of Coherently Controlled Photocurrent in Unbiased, Bulk GaAs. *Phys. Rev. Lett.* **78**, 306–309 (1997).
30. Zhao, H., Loren, E. J., Van Driel, H. M. & Smirl, A. L. Coherence Control of Hall Charge and Spin Currents. *Phys. Rev. Lett.* **96**, 246601 (2006).
31. Sederberg, S. et al. Vectorized optoelectronic control and metrology in a semiconductor. *Nat. Photon.* **14**, 680–685 (2020).

We have also quoted the following papers on the studies of the terahertz radiations by two-color quantum interference control in conventional semiconductors.

32. Costa, L., Betz, M., Spasenović, M., Bristow, A. D. & Van Driel, H. M. All-optical injection of ballistic electrical currents in unbiased silicon. *Nat. Phys.* **3**, 632–635 (2007).
33. Spasenović, M., Betz, M., Costa, L. & Van Driel, H. M. All-optical coherent control of electrical currents in centrosymmetric semiconductors. *Phys. Rev. B* **77**, 085201 (2008).
34. Sames, C., Ménard, J. M., Betz, M., Smirl, A. L. & Van Driel, H. M. All-optical coherently controlled terahertz ac charge currents from excitons in semiconductors. *Phys. Rev. B* **79**, 045208 (2009).

To refer to the previous theoretical study focusing on the compound semiconductors, which predicted that charge currents would be generated when two excitons with different symmetry are simultaneously excited and that the phase of the current could be changed by varying the phase difference between the polarizations of two excitons, we have added the following sentences.

[Line 4 to 8 of page 4]

“The previous theoretical study focusing on the compound semiconductors predicted that charge current would be generated when two excitons with different symmetry are simultaneously excited and that the phase of the current could be changed by varying the phase difference between the polarizations of two excitons⁴⁴. In this case, the terahertz radiation and its phase control might be possible.”

In this revision, we have quoted the previous theoretical study as follows.

44. Marti, D. H., Dupertuis, M. A. & Deveaud, B. Optical injection of charge current in quantum wires: Oscillations induced by excitonic effects. *Phys. Rev. B*, **72**, 075357 (2005).

To highlight the important achievements and findings in our study, we have added the following descriptions in the introductory part.

[Line 1 to 15 of page 6]

“In the following, we report that in $[\text{Ni}(\text{chxn})_2\text{Br}]\text{Br}_2$, the amplitude and phase of radiated terahertz waves can be controlled independently; by setting the appropriate value of the interval between the incident $\omega -$ and $2\omega -$ pulses, that is, the creation time difference of two excitons with the accuracy of ten attoseconds, we can select the continuous amplitude control of the terahertz radiation with its phase fixed to $\pm\pi$ or the continuous phase control from $-\pi$ to $+\pi$ with its amplitude fixed. We also demonstrate that the central frequency of the terahertz radiation can be tuned by adjusting the creation-time difference of two excitons. Moreover, we show that the efficiency of the terahertz radiation in $[\text{Ni}(\text{chxn})_2\text{Br}]\text{Br}_2$ is much higher than those in conventional semiconductors, Si and Ge, in the similar framework of the third-order optical nonlinearity. The efficiency of the terahertz radiation in $[\text{Ni}(\text{chxn})_2\text{Br}]\text{Br}_2$ is also higher than that from ZnTe originating from the general OR framework based upon the second-order optical nonlinearity. In the final part, we discuss the phase relaxation dynamics of excitons in $[\text{Ni}(\text{chxn})_2\text{Br}]\text{Br}_2$ by analyzing the dependence of the terahertz radiation intensity on the creation time difference of even-parity and odd-parity excitons.”

In addition, to emphasize the uniqueness of the controllability of electric-field amplitude and phase demonstrated in our study, we have added the following sentences.
[Line 6 to 10 of page 8]

“Thus, by selecting the appropriate value of Δt , we can achieve two types of controls of amplitude and phase of terahertz radiations; the continuous control of the electric-field amplitude with the electric-field phase fixed to $\pm\pi$ or the continuous control of the electric-field phase from $-\pi$ to $+\pi$ with the electric-field amplitude fixed.”

To compare the terahertz radiation phenomena between the 1D Mott insulator and conventional semiconductors, we have added a new subsection in the Results section, **“Comparison of terahertz radiations from $[\text{Ni}(\text{chxn})_2\text{Br}]\text{Br}_2$ and conventional semiconductors”**.

The descriptions about the comparison of the terahertz radiation in $[\text{Ni}(\text{chxn})_2\text{Br}]\text{Br}_2$ with those from CdSe and CdTe are extracted below.

[Line 17 of page 11 to line 12 of page 12]

“In CdSe and CdTe, it has been attempted to generate the phase-controlled terahertz pulses by the excitations of $1s$ and $2p$ excitons with different symmetries³⁴. However, the exciton binding energies of these compound semiconductors are still small; the binding energy of the lowest exciton is approximately 15 meV in CdSe and 11 meV in CdTe. In those compounds, therefore, it is difficult to resonantly excite only those excitons and the generated terahertz waves necessarily include two types of the components, which originate from the interband transitions and from the exciton transitions. As a result, when varying the phase difference between the $\omega -$ and $2\omega -$ pulses, both amplitude and phase of terahertz radiation are simultaneously changed, and the continuous control of the electric-field phase of the terahertz radiation with the electric-field amplitude fixed, as demonstrated in $[\text{Ni}(\text{chxn})_2\text{Br}]\text{Br}_2$, had not been achieved^{32,34}. In addition, these compounds are three-dimensional (3D) semiconductors and the oscillator strength of the lowest exciton is not so large. Therefore, the intensity of the terahertz radiation caused by the exciton transitions is much smaller than that caused by the interband transitions. In our study, in $[\text{Ni}(\text{chxn})_2\text{Br}]\text{Br}_2$, the terahertz radiation occurs only via the quantum interference between the odd-parity and even-parity excitons and is not affected by the interband transitions. It is because the binding energies of the odd-parity exciton and even-parity exciton, which are approximately 90 meV and 80 meV, respectively⁴⁰, are sufficiently large as compared to the spectral width of the excitation pulses (approximately 20 meV) and the oscillation strength is concentrated on the lowest odd-parity exciton reflecting the 1D nature of the electronic system.”

The physical mechanism for the unique controllability of amplitude and phase of terahertz radiation in $[\text{Ni}(\text{chxn})_2\text{Br}]\text{Br}_2$ demonstrated in our study is explained in the Discussion section from line 6 of page 14 to line 17 of page 15 in the revised MS (from line 11 of page 8 to line 21 to page 9 in the previous MS).

Comment (2) of Reviewer #2

The authors state that the phase and frequency control “are challenging to obtain in the optical rectification (OR) framework since the OR process is a simple classical electromagnetic phenomenon.”

I do not agree with this. All the nonlinear responses including OR is the quantum mechanical phenomenon in the microscopic point of view, and one can just also recognize OR in a classical phenomenological picture. THz emission by OR is a kind of differential frequency generation (DFG) process. DFG using two different pulses can also tune both the center frequency and the CEP of THz emission by adjusting the time delay between the two pulses. See e.g., Opt. Lett. 33, 2767 (2008), Nature Photon. 8, 119 (2014), Opt. Lett. 42, 129 (2017).

Reply to comment (2) of Reviewer #2

We agree with the opinion of Reviewer #2. A general way to generate a terahertz electromagnetic wave is to use the optical rectification that occurs when a femtosecond laser pulse is incident to a second-order nonlinear optical crystal with no inversion symmetry, such as ZnTe. In this case, a terahertz wave is generated by injecting a pulse with the photon energy lower than the band gap of the crystal, so that this process is sometimes treated as a classical electromagnetic effect by regarding $\chi^{(2)}$ as an optical constant without considering its quantum mechanical origin. However, as Reviewer #2 pointed out, this process uses electronic transition on the higher energy side, so it should be treated quantum mechanically, and $\chi^{(2)}$ can also be expressed using the electronic transition. Therefore, we have deleted the following phrase in the previous MS.

“... since the OR process is a simple classical electromagnetic phenomenon.”

We have also deleted the following statements from line 19 of page 6 to line 5 of page 7 in the previous MS since the method to control the phases of terahertz waves explained there uses the special optical devices and therefore is not directly related to the content of our paper.

“Until now, controlling the phase of terahertz waves has been done in two steps. First,

terahertz waves are generated via second-order optical nonlinearity by injecting a femtosecond laser pulse into a nonlinear optical crystal with no inversion symmetry, and then they are passed through a specially designed optical element with a large energy loss^{43,44}. In that optical element, the effect of simple classical electromagnetism is used, i.e., the velocity of the terahertz wave changes in dependence on polarization. Our method, on the other hand, is quite different from previous methods. It uses the quantum mechanical interference effect between nearly degenerate odd-parity and even-parity excitons that are unique to 1D Mott insulators. This method is extremely superior in that it achieves the emission of terahertz wave by simultaneously generating odd-parity and even-parity excitons, while it allows to freely control its phase ϕ only by tuning Δt .”

By making the above correction, we have removed the following references in the new MS.

[43 in the previous MS] Nagai, M. et al. Achromatic THz wave plate composed of stacked parallel metal plates. *Opt. Lett.* **39**, 146–149 (2014).

[44 in the previous MS] Kawada, Y., Yasuda, T. & Takahashi H. Carrier envelope phase shifter for broadband terahertz pulses. *Opt. Lett.* **41**, 986–989 (2016).

As suggested by Reviewer #2, the previous studies have reported a method of generating a mid-infrared pulse from two near-infrared pulses with different frequencies via a differential frequency generation (DFG) process in a second-order nonlinear optical crystal [A. Sell et al. *Opt. Lett.* **33**, 2767 (2008), O. Schubert et al. *Nat. Photon.* **8**, 119 (2014), B. Liu et al. *Opt. Lett.* **42**, 129 (2017)]. In those reports, it has been shown that by adjusting the time difference between the incident two pulses, both the central frequency and carrier envelope phase (CEP) of the mid-infrared pulse can be controlled. Using the same method, it may be possible to generate a terahertz pulse and control its frequency and phase. However, when a pulse with a temporal width of approximately 100 fs is incident to a second-order nonlinear optical crystal, a terahertz pulse is efficiently emitted via the intra-pulse OR process and its phase and frequency is difficult to change. This process is expected to be more effective than the two-pulse OR process due to the third-order optical nonlinearity. In addition, the experimental setup for the DFG using two different near-infrared pulses is somewhat complicated and requires a feedback mechanism to stabilize the optical path lengths of the two near-infrared pulses. In contrast, in the method using the third-order optical nonlinearity, the frequencies of the two pulses are ω and 2ω , so that we can easily prepare a 2ω –pulse from an ω –pulse via the SHG process. Therefore, in this method, we can generate a phase-stable terahertz pulse

using a simple optical setup and control its amplitude and phase by changing the time difference of ω – and 2ω – pulses as mentioned above. This point is discussed in the new subsection in the Results section, “**Comparison of terahertz radiations from [Ni(chxn)₂Br]Br₂ and conventional semiconductors**”.

The discussion related to the comment of Reviewer #2 is shown below.

[Line 3 to 16 of page 13]

“Controlling the phase and frequency of a terahertz pulse might be possible using a second-order nonlinear optical effect. In fact, it was reported that a mid-infrared pulse can be generated from two near-infrared pulses with different frequencies via the differential frequency generation (DFG) process^{53–55}. In those studies, the tuning of phase and frequency of the mid-infrared pulse was achieved by adjusting the time difference between two near-infrared pulses. However, this technique would be somewhat difficult to apply for a terahertz radiation. It is because a terahertz radiation due to the simple single-pulse OR process occurs by each incident pulse, making it difficult to control the phase and frequency of the terahertz radiation via the DFG process. In addition, in the DFG method, a complicated measurement system was necessary to prepare two optical paths with precisely adjusted lengths for two different near-infrared pulses. In contrast, the method using the third-order optical nonlinearity reported here requires only two pulses with frequencies of ω and 2ω . The latter pulse can be easily obtained from the former pulse using the SHG process within the same optical path (see Supplementary Fig. 1a), so the measurement system is much more simplified.”

In this discussion, we have quoted the following references.

53. Sell, A., Leitenstorfer, A. & Huber, R. Phase-locked generation and field-resolved detection of widely tunable terahertz pulses with amplitudes exceeding 100 MV/cm. *Opt. Lett.* **33**, 2767–2769 (2008).
54. Schubert, O. et al. Sub-cycle control of terahertz high-harmonic generation by dynamical Bloch oscillations. *Nat. Photon.* **8**, 119–123 (2014).
55. Liu, B. et al. Generation of narrowband, high-intensity, carrier-envelope phase-stable pulses tunable between 4 and 18 THz. *Opt. Lett.* **42**, 129–131 (2017).

Comment (3) of Reviewer #2

The main text begins with the interest in control of solids by using intense THz pulse. In the abstract, the authors state that “an intense terahertz wave is generated.” Nevertheless, I could not find any value of the THz field strength achieved in this work.

In general, the third-order nonlinearity is weaker than the conventional second-order response, and therefore how strong field is available can be the disadvantage for this work. The authors should show the peak value of the THz field.

Reply to comment (3) of Reviewer #2

As Reviewer #2 pointed out, it is important to estimate the energy efficiency of the terahertz radiation and the absolute value of its electric-field amplitude. We have performed the estimations of those values and detailed in the new MS. We have also measured the terahertz radiations from Si and Ge in the same experimental setup as that used in the terahertz radiation measurement of $[\text{Ni}(\text{chxn})_2\text{Br}]\text{Br}_2$. The results showed that in $[\text{Ni}(\text{chxn})_2\text{Br}]\text{Br}_2$, the efficiency of the terahertz radiation per unit length is orders of magnitude higher than those in conventional semiconductors. These results are presented in the new subsection in the Results section, “**Comparison of terahertz radiations from $[\text{Ni}(\text{chxn})_2\text{Br}]\text{Br}_2$ and conventional semiconductors**” in the revised MS.

The descriptions about the comparison of the terahertz radiation efficiency in $[\text{Ni}(\text{chxn})_2\text{Br}]\text{Br}_2$ with those from Si and Ge are extracted below.

[Line 16 of page 9 to 16 of page 11]

“As mentioned in the introductory part, in conventional semiconductors with inversion symmetry such as Si, Ge, InP, and GaAs, terahertz radiations based upon the third-order nonlinear optical process expressed by $P(0) =$

$$\frac{3}{4}\varepsilon_0\chi^{(3)}(0; \omega, \omega, -2\omega)E(\omega)E(\omega)E(-2\omega)$$

using $\omega -$ and $2\omega -$ pulses have also

been reported. In those semiconductors, the two-photon excitation by the $\omega -$ pulse and one-photon excitation by the $2\omega -$ pulse are both due to the interband transitions. It is interesting to compare the efficiency of the terahertz radiation from $[\text{Ni}(\text{chxn})_2\text{Br}]\text{Br}_2$ with those from those semiconductors, which originate from the similar third-order nonlinear optical processes. For this subject, we have measured the terahertz radiations from $[\text{Ni}(\text{chxn})_2\text{Br}]\text{Br}_2$, Si, and Ge with two incident pulses of $\hbar\omega = 0.64$ eV and $2\hbar\omega = 1.28$ eV in the same experimental setup with the reflection configuration (see Supplementary Note 1). For Si and Ge, crystals cut out of the (100) plane were used. The excitation fluences of the $\omega -$ and $2\omega -$ pulses are 1.77 mJ/cm² and 0.11 mJ/cm², respectively. The $\omega -$ pulse was used for the two-photon excitation, and its fluence was much higher than that of the $2\omega -$ pulse used for the one-photon excitation.

In Fig. 6, the electric field waveforms of the terahertz radiations from Si and Ge are shown by the green and red lines, respectively, together with that emitted from $[\text{Ni}(\text{chxn})_2\text{Br}]\text{Br}_2$ (the black line). The electric field amplitude (intensity) of the terahertz

radiation from $[\text{Ni}(\text{chxn})_2\text{Br}]\text{Br}_2$ is approximately 2.4 (5.8) times higher than that from Ge and 50 (2500) times higher than that from Si. The terahertz radiation from Si is much smaller than that from Ge. It is because the direct interband transition cannot be excited by the 1.28 eV pulse in Si but can be excited in Ge.

Next, we compare the terahertz radiation efficiency of $[\text{Ni}(\text{chxn})_2\text{Br}]\text{Br}_2$ and Ge more precisely. In our experiments, the polarizations of ω – and 2ω – pulses are orthogonal (see Supplementary Fig. 1b). In the experiments on $[\text{Ni}(\text{chxn})_2\text{Br}]\text{Br}_2$, they are tilted by ± 45 degrees with respect to the 1D Ni chain (see Supplementary Note 1). If both polarizations are parallel to the 1D Ni chain (the b -axis) while maintaining the intensities of two pulses, the electric field amplitude of the terahertz radiation is expected to increase by a factor of $2\sqrt{2}$ and its intensity by a factor of 8. On the other hand, in Ge, the terahertz radiation intensity for the polarizations of two excitation pulses parallel to each other is roughly 1.1 times higher than that for their polarizations perpendicular to each other³³. Therefore, the intensity of the terahertz radiation from $[\text{Ni}(\text{chxn})_2\text{Br}]\text{Br}_2$ is estimated to be approximately 42 times ($= 5.8 \times 8 / 1.1$) as large as that from Ge.

More strictly, it is necessary to discuss the terahertz radiation efficiency by taking the penetration depths of the incident pulses into account. Since the penetration depth of the 2ω – pulse, $l(2\omega)$, is much shorter than that of the ω – pulse, $l(\omega)$, the terahertz radiation is emitted only from a region as deep as $l(2\omega)$. Therefore, it is reasonable to consider that an electric-field amplitude of the terahertz radiation is proportional to $l(2\omega)$ and its intensity is proportional to $[l(2\omega)]^2$. $l(2\omega)$ is approximately 20 nm in $[\text{Ni}(\text{chxn})_2\text{Br}]\text{Br}_2$ and approximately 500 nm in Ge⁵⁰. To compare the terahertz radiation efficiency per unit length between $[\text{Ni}(\text{chxn})_2\text{Br}]\text{Br}_2$ and Ge, it is necessary to multiply the terahertz radiation intensity in the former approximately 600 times. Taking this factor into account, the terahertz radiation efficiency per unit length in $[\text{Ni}(\text{chxn})_2\text{Br}]\text{Br}_2$ is estimated to be 2.6×10^4 times as large as that of Ge. This extremely high efficiency of the terahertz radiation in $[\text{Ni}(\text{chxn})_2\text{Br}]\text{Br}_2$ is characteristic of 1D Mott insulators, in which third-order nonlinear optical responses associated with nearly degenerate odd- and even-parity excitons are very large. Such large third-order nonlinear optical responses originate from the effects of strong electron correlation and one-dimensionality^{36–38}.”

It is interesting to compare the terahertz radiation efficiency from $[\text{Ni}(\text{chxn})_2\text{Br}]\text{Br}_2$ via the third-order optical nonlinearity (two-pulse OR) with that from ZnTe via the second-order optical nonlinearity (single-pulse OR). Our additional measurements revealed that the terahertz radiation from $[\text{Ni}(\text{chxn})_2\text{Br}]\text{Br}_2$ was stronger than that from ZnTe. Although the difference is that the former used resonant exciton excitations and the

latter used non-resonant excitations in the transparent region, this result indicates that the efficiency of the terahertz radiation from the two-pulse OR in $[\text{Ni}(\text{chxn})_2\text{Br}]\text{Br}_2$ is very high.

We have also reported this result in the new subsection “**Comparison of terahertz radiations from $[\text{Ni}(\text{chxn})_2\text{Br}]\text{Br}_2$ and conventional semiconductors**” in the Results section.

The discussion about this result is extracted below.

[Line 13 of page 12 to line 2 of page 13]

“Using the same experimental setup, we also measured the terahertz radiation from ZnTe, which is a second-order nonlinear optical crystal without inversion symmetry and widely used to generate terahertz electromagnetic waves via the OR process described by $P(0) = 2\varepsilon_0\chi^{(3)}(0; \omega, -\omega)E(\omega)E(-\omega)^{51,52}$. In general, the third-order nonlinear optical effect described by $P(0) = \frac{3}{4}\varepsilon_0\chi^{(3)}(0; \omega, \omega, -2\omega)E(\omega)E(\omega)E(-2\omega)$ should be much smaller than the second-order nonlinear optical effect. Since the excitation fluence of the ω –pulse is approximately 16 times higher than that of the 2ω –pulse, it can be considered that the terahertz radiation occurs via the OR induced by the ω –pulse in ZnTe. The electric field waveform of the terahertz radiation from ZnTe with the (110) plane cut out is shown by the blue line in Fig. 6. The electric field amplitude of the terahertz radiation emitted from $[\text{Ni}(\text{chxn})_2\text{Br}]\text{Br}_2$ is 5.4 times as large as that from ZnTe. Although the difference is that the former used resonant exciton excitations and the latter used non-resonant excitations in the transparent region, this result also indicates that the efficiency of the terahertz radiation from $[\text{Ni}(\text{chxn})_2\text{Br}]\text{Br}_2$ is very high.”

We have also added the related references in the revised MS.

51. Nahata, A., Weling, A. S. & Heinz, T. F. A wideband coherent terahertz spectroscopy system using optical rectification and electro - optic sampling. *Appl. Phys. Lett.* **69**, 2321–2323 (1996).
52. Blanchard, F. et al. Generation of 1.5 μJ single-cycle terahertz pulses by optical rectification from a large aperture ZnTe crystal. *Opt. Express* **15**, 13212–13220 (2007).

As Reviewer #2 pointed out, the description of “an intense terahertz wave is generated.” in the abstract might be misleading. Therefore, we have modified this part as follows.

[In the abstract of the previous MS]

“Here, we show that in a one-dimensional Mott insulator of a nickel-bromine chain

compound an intense terahertz wave is generated via strong electron modulations due to quantum interference between odd-parity and even-parity excitons produced by two-color femtosecond pulses.”

[In the abstract of the new MS]

“Here, we show that in a one-dimensional Mott insulator of a nickel-bromine chain compound a terahertz wave is generated with high efficiency via strong electron modulations due to quantum interference between odd-parity and even-parity excitons produced by two-color femtosecond pulses.”

Comment (4) of Reviewer #2

In the abstract, the authors also state that “one can control all of the phase, frequency, and amplitude of terahertz waves by adjusting the creation-time difference of two excitons”. Figure 5b shows that the peak of the Fourier-amplitude spectra shifts from 1.2 to 1.5 THz. However, the data are normalized, and the bumpy spectral shape for the weakened condition looks like that the THz output is just suppressed due to a deconstructive interference. “Control of frequency” sounds overstatement.

Reply to comment (4) of Reviewer #2

We would like to thank Reviewer #2 for his/her valuable comment. This frequency shift of the terahertz radiation with increase of Δt is an important feature in our method using well-defined two exciton levels with different energies. In the interband transitions in conventional semiconductors, it is reasonable to consider that both of odd-parity and even-parity excited states exist continuously above the band edge. In those cases, no energy differences exist between odd and even excited states and, therefore, we can expect no frequency shift of the terahertz radiation by changing the time difference Δt between the ω – and 2ω – pulses. On the other hand, in the case of the 1D Mott insulator of $[\text{Ni}(\text{chxn})_2\text{Br}]\text{Br}_2$, the excited states involved in the terahertz radiation are the well-defined even- and odd-parity excitonic states. Therefore, when the delay time Δt of the 2ω – pulse relative to the ω – pulse is increased, only the process in the lefthand side shown in Fig. 4c becomes active. This process is also shown in Fig. 4d. The frequency of the terahertz radiation in this case should approach 10 meV (2.5 THz), which is the energy difference between the even- and odd-parity excitonic states.

In our study, we used a nonlinear optical crystal of ZnTe with the thickness of 1 mm in the electro-optical sampling to measure the electric field of the terahertz radiation. In this case, as the frequency increases from the low frequency side, the detection sensitivity

of the electric field increases, having a maximum around 1 THz, and then sharply decreases, being very small at 2.5 THz. For this frequency dependence of the detection sensitivity, the high frequency shift up to 2.5 THz was difficult to observe. However, the high frequency shift with increasing Δt is characteristic of the terahertz radiation phenomenon based upon the third-order optical nonlinearity associated with even-parity and odd-parity exciton states having *different frequencies*. In the 1D Mott insulators of $[\text{Ni}(\text{chxn})_2\text{X}]Y_2$, the energy difference $\hbar(\omega_e - \omega_o)$ between the even-parity and odd-parity excitonic states can be systematically changed by changing bridging halogen ion X and counter ion Y [M. Ono et al. *Phys. Rev. B* **70**, 085101 (2004).], which ranges from 10 meV to 90 meV. Thus, it is possible to vary the frequency of the terahertz radiation over the wider frequency range by selecting appropriately a temporal width of the incident laser pulses, a material with the desired energy of $\hbar(\omega_e - \omega_o)$, and a frequency range of the terahertz radiation detectable.

To emphasize this feature in 1D Mott insulators of halogen-bridged nickel-chain compounds, we have added the following sentences.

[Line 18 of page 15 to 4 of page 16]

“Such a high-frequency shift with increasing Δt is characteristic of the terahertz radiation due to the third-order optical nonlinearity associated with well-defined even-parity and odd-parity excitonic states with different energies. Such a shift was not observed in the similar terahertz radiations in conventional 3D semiconductors, where excitonic effect is very weak. In the series of halogen-bridged nickel-chain compounds expressed as $[\text{Ni}(\text{chxn})_2\text{X}]Y_2$, the energy difference between the even-parity and odd-parity excitons, $\hbar(\omega_e - \omega_o)$, can be widely changed from 10 meV to 90 meV by the choice of bridging halogen ion X and counterion Y ³⁸. Therefore, if we appropriately select a temporal width of the incident laser pulses, a material with the desired energy of $\hbar(\omega_e - \omega_o)$, and a frequency range of the terahertz radiation detectable, it would be possible to control the terahertz radiation frequency over a wider range by varying Δt .”

REVIEWER COMMENTS

Reviewer #1 (Remarks to the Author):

The authors have conducted a thorough revision of their manuscript, which resulted in a substantial improvement. The additional experiments as well as the revised text made the work more convincing. The authors have addressed my first comment regarding the optical-to-THz power conversion efficiency and also provided satisfying estimates of the THz peak electric field of the generated THz pulses. However, I still have few minor remarks that I hope the authors address in the revised manuscript, without any doubt that this work is worth publishing in Nature Communications.

1) As far as I understand, the authors estimated the THz electric field strength from the electro-optic sampling, as described in the Methods, and consequently calculated the corresponding intensity via the integral at the end of that section. From the calculated intensity, I guess that the authors estimated the corresponding THz energy of $\sim 10^{-13}$ J. I do not think that this very low energy is measurable, but it is obviously estimated. Whether estimated or measured, I hope that the authors mention explicitly in the manuscript how this magnitude was obtained. I further suggest numbering the equations in the Methods.

2) I am curious to know the damage threshold of the used Mott insulator. This will additionally provide a fair comparison with the other materials investigated in this work. I think that a summary table listing the investigated materials and the various aspects of the comparison will be a good addition, but it is optional.

3) Regarding the matching of colors of the curves and legends of Fig. 5h, especially for the green and blue, I still do not think they really match, but I will leave this to be judged by the production team of Nature Communications.

Reviewer #2 (Remarks to the Author):

Adding new references, the authors revised the manuscript and it reads much better. However, the main message of the paper is unchanged. Again, I do not feel that the result represents a breakthrough or an important step in terahertz emission, light-matter interaction, or control of matter because essentially same phenomena have been reported in the literatures. The authors summarized the difference from previous studies as well-defined parities of excitons, high efficiency of terahertz radiation, and determination of phase relaxation time of excitons. But all the things look a minor extension. The high efficiency is a comparison among the methods using third-order nonlinearity, and the difference from conventional semiconductors is just a factor of 3 or 4. The obtained field is only 100 V/cm, which is much weaker than usual terahertz emission by typical second-order process. The authors expect a frequency tenability for the use of other compounds, but it is not demonstrated in this work.

I cannot recommend it for the publication in Nature Communication. I suggest Communication Physics as a better outlet for this work

Report on the Manuscript, “Terahertz radiation by quantum interference of excitons in a one-dimensional Mott insulator”, by Tatsuya Miyamoto *et al.*, ; Ref: NCOMMS-22-38702A

The manuscript deals with experimental demonstration of effective generation of terahertz waves from a typical 1D Mott insulator $[\text{Ni}(\text{chxn})_2\text{Br}]\text{Br}_2$. They claim to achieve this by exciting odd-parity, one-photon state and even-parity, two photon state using two-color femtosecond pulses and based on the quantum interference of the two third-order nonlinear optical responses associated with terahertz radiation, which are characterized by $\chi^{(3)}(-\omega_{\text{THz}}; \omega, \omega, -(2\omega - \omega_{\text{THz}}))$ and $\chi^{(3)}(-\omega_{\text{THz}}; -\omega, -\omega, 2\omega + \omega_{\text{THz}})$. The two nonlinear processes interfere, and the radiation intensity oscillates depending on the delay time of generation of the above mentioned two low energy excitations. The authors have discussed the physical mechanism of the observed terahertz radiation and the Δt dependence of its phase, amplitude, and frequency in the third-order optical nonlinearity framework. It was shown that the phase, amplitude, and frequency of the terahertz waves can be controlled by the fine tuning of the generation time difference of two excitons. They have estimated the energy conversion efficiency from two excitation pulses to a terahertz pulse in $[\text{Ni}(\text{chxn})_2\text{Br}]\text{Br}_2$ with their method using the third-order optical nonlinearity. The electric field amplitude of the terahertz radiation emitted from $[\text{Ni}(\text{chxn})_2\text{Br}]\text{Br}_2$ is found to be 5.4 times as large as that from ZnTe. In general, the finding of the work is very interesting and would be of interest to the large community of researchers working in theoretical, experimental aspects of non-linear optics of strongly correlated systems. Thus it may deserve publication. The current reviewer wants to draw attention to the authors on the following aspects.

1. The way the manuscript is written it would appear to the readers till a large part of the manuscript that it is only a proposal and the authors are probably providing a fitting estimate(s). The authors should claim/mention in the introduction itself clearly that this work is an experimental demonstration of efficient terahertz generation for strongly correlated material the physics of which is under active research of condensed matter physics community. In order to establish so what is achieved in which section/subsection must be mentioned at the end of the Introduction itself.
2. Experimental findings may be correct but the physical explanations the authors provided and the model used for the support for explanation in the first few pages of the manuscript is questionable. It appears as if it is the authors who are first time proposing the terahertz generation in one dimensional strongly correlated quantum wire. This is not true (see below). In the process the authors ignore many important literatures and lose to attract general nature of their work in the subject.

Gigantic optical nonlinearities are observed in various one dimensional Mott-Hubbard insulators represented by solid circles compared to other one dimensional materials (hollow circles). Experimental and theoretical research predicts occurrence of nearly degenerate one and two photon states in these one dimensional Mott-Hubbard insulators which are very strongly dipole coupled. This is the primary reason for huge ground state optical non-linearities like TPA, THG, EA. This story began in the year 2000 (Nature 405, 929–932 (2000), Physical Review Letters 85, 2204, (2000)).

Subsequently it was realized that not only the lowest one/two photon states are nearly degenerate but there would be similar higher energy two photon states in the electronic structure. ``*Excited state nonlinear optics of quasi-one-dimensional Mott-Hubbard insulators*”, Physical Review B **75**, 235127, 2007 (and references therein). Based on exact numerical results on finite size one-dimensional Mott-Hubbard insulators as well as available experimental data it was already predicted to be a good source of terahertz radiation. Further predications with explicit calculations were published later in, “*Gigantic stimulated Raman scattering in one-dimensional Mott-Hubbard insulators: A possible THz source*” Physics Letters A **374**, 2379 (2010); <https://doi.org/10.1016/j.physleta.2010.03.055> IF 2.654 ISSN 0375-9601. Both ground and excited state optical nonlinearities are required for efficient terahertz communication medium. “*Gaint ground and excited state optical non-linearty in one dimensional cuprates*”, Europhysics Letters **75**, 468 (2006); <https://doi.org/10.1209/epl/i2006-10337-8>. Question is, is the strong electron-electron repulsion (U) and/or confinement enough for terahertz communication source!?! Then how to explain the material dependent properties presented in the figure! Does the holon-doublon model have enough! The reviewer requests authors’ attention to these facts! The smallest parameter in the game is the presence of “Br” atoms (its site energy, the electronegativity, note the orders of magnitude difference in $\text{Im } \chi^3$ in the figure even though they belong to the same class). Clearly, authors themselves were in doubt (/uncomfortable) in page 5, line 15.

The authors asserted that, “The intense terahertz pulse has been used as an excitation pulse to control the electronic state properties such as ..., superconductivity9–11, magnetism19–23...” It is well known that superconductivity is realized only with varied oxygen content (in cuprates); magnetism is also controlled through oxygen content. . Large J ~2800 K of Ni-Br-Br as compared with that of Ni-Cl-Cl and Ni-Cl-NO₃ is attributable to the smaller value of charge transfer gap (difference in site energies of Ni and X (= Br / Cl ...)). The difference of J between Ni-Cl-Cl; J ~2200 K and Ni-Cl-NO₃ J ~2100 K has the same origin); U or V is same or similar in all these compounds. (See for example, PHYSICAL REVIEW B **70**, 085101 (2004)). There would be no exchange coupling without intervening legend atoms (Br, O etc.). In short, the absence of Br atoms

in holon-doublon model is not a practical one as far as its application to non-linear optical medium required for terahertz communication is concerned. The authors use over simplified cartoon pictures which are arguable; For example, In Fig. 1a if the states $|L\rangle$ and $|R\rangle$ are separated from the ground state by U then so is their linear combinations $\phi_e \pm \phi_o$. Thus it cannot describe correctly the location (energetically) of the one/two-photon states in Fig 2c. The difference in 1-/2-photon states in case of Sr2CuO3 is much smaller as compared to those in the Ni-Cl-Cl and Ni-Cl-NO3 but still Sr2CuO3 has much lower $\text{Im } \chi^3$ (cf. the figure). In case of Ca2CuO3 the energy orderings of the 1-ph/2-ph states are reversed. The holon-doublon model does not have enough ingredients to capture the essential physics of the whole classes of strongly correlated materials.

The appropriate model required for 1-d Mott-Hubbard insulators, in the present purpose is (Physics Letters A 374, 2379 (2010)),

$$H = t_{pd} \sum_{(ij),\sigma} (d_{i\sigma}^\dagger p_{j\sigma} + \text{h.c.}) + \varepsilon_d \sum_{i,\sigma} n_i^d - \varepsilon_p \sum_{i,\sigma} n_i^p \\ + \sum_i U_{dd} n_{i\uparrow}^d n_{i\downarrow}^d + \sum_i U_{pp} n_{i\uparrow}^p n_{i\downarrow}^p + \sum_{i,\delta} V_\delta n_i n_{i+\delta}$$

The first term represents hopping probability between the Cu (Ni) and O (X) sites, the second and third terms represent respectively the on-site energies of the Cu (Ni) and O (X), thereby defining the charge transfer energy $2\varepsilon = \varepsilon_{\text{O(X)}} - \varepsilon_{\text{Cu(Ni)}}$. The fourth and the fifth terms represent respectively the onsite Coulomb repulsions, i.e., the repulsion energy felt by two electrons occupying the same Cu (Ni)-3d or O (X)-p orbital with opposite spins. The last term is known as intersite Coulomb repulsion (V), repulsive energy felt by two electrons occupying the neighboring Cu (Ni) and O (X) (second neighbor Cu (Ni) and Cu (Ni), etc.) sites. In fact these materials under considerations are more of charge transfer Mott insulators ... see for example, "Origin of Giant Optical Nonlinearity in Charge-Transfer-Mott Insulators: A New Paradigm for Nonlinear Optics", Phys. Rev. Lett. **86**, 2086 (2001). Furthermore indeed two band model is required to explain the observed large non-linearity and ultrafast recovery pre-requisite for THz communications; see for example, "Interband two-photon transition in Mott insulator as a new mechanism for ultrafast optical non-linearity", Int. Jr. Mod. Phys. B15,3628(2001); <https://doi.org/10.1142/S0217979201008305> (also, Book Chapter on "Excitonic processes in condensed matter", Edited by Kikuo Cho, Atsuo Matsui, World Scientific, ISBN 981-02-4588-2. Page no. 60 (2001)). Since the authors are NOT directly using the holon-doublon model for experimental demonstration of Thz radiation, they should use a practical model.

The used model in general perhaps cannot account for all the materials in the category of strongly correlated natural quantum wires; for example in case of Ca2CuO3, the two photon state is below the one photon state. Possibly further, in that case $\chi^{(3)}(-\omega_{\text{THz}}; \omega, \omega, -(2\omega - \omega_{\text{THz}}))$ may not remain the dominant mechanism (authors should comment on that).

It is well known that the low energy excitations in 1-d strongly correlated systems are gapless spin excitations. Thus, authors should provide an estimate of "the delay of the

generation time of the odd-parity excitons relative to the even-parity excitons” in contrast to the decay time to the ground state. The gapless spin excited states are very similar in nature to the Antiferromagnetic ground state and being gapless it is expected that some of the population generation will also take place from these states. Authors should comment on such and other competitive processes.

Summary of changes

We have thoroughly revised the manuscript (MS).

- (1) Taking comment (1) of Reviewer #1 into account, we have revised the related description in the main paper to make it clear that the energy of the emitted terahertz wave was estimated from the terahertz electric field waveform quantitatively measured. In addition, we numbered the equations in the Methods.
- (2) Taking comment (2) of Reviewer #1 into account, we have added the values of the damage thresholds of the bromine-bridged nickel-chain compound and germanium.
- (3) Taking comment (3) of Reviewer #1 into account, we have modified the colors of the curves in Fig. 5h.
- (4) Taking comment (1) of Reviewer #3 into account, we have modified the sentence in the Introduction to make it clear that this study is an experimental demonstration of efficient terahertz generation from 1D Mott insulators and that their physics has been actively studied in the condensed matter physics community.
- (5) Taking comment (2-a) of Reviewer #3 into account, we have added the sentences to the main paper showing that theoretical predictions for efficient terahertz radiations in 1D Mott insulators have been previously proposed and that the mechanism of the terahertz radiation proposed in our study is different from those previously proposed.
- (6) Taking comment (2-b) of Reviewer #3 into account, we have revised the explanation in the Introduction to show that the mechanism of terahertz radiation proposed in this study holds for the two-band Hubbard system. In connection with this revision, we have revised Fig. 1 in the revised MS and deleted Fig. 2a in the old MS.
- (7) Taking comment (2-d) of Reviewer #3 into account, we have modified the descriptions to clearly state the fact that the relaxation time of excitons to the ground state is sufficiently long compared to their phase relaxation time.
- (8) We have added several new references necessary for the revisions shown above and renumbered some references in the revised MS.

All the changed parts are marked in yellow in the additional file named Marked MS2.pdf.

Comment (0) of Reviewer #1

The authors have conducted a thorough revision of their manuscript, which resulted in a substantial improvement. The additional experiments as well as the revised text made the work more convincing. The authors have addressed my first comment regarding the optical-to-THz power conversion efficiency and also provided satisfying estimates of the THz peak electric field of the generated THz pulses. However, I still have few minor remarks that I hope the authors address in the revised manuscript, without any doubt that this work is worth publishing in Nature Communications.

Reply to comment (0) of Reviewer #1

We would like to appreciate his/her comment; “The additional experiments as well as the revised text made the work more convincing.” We show the replies to his/her comments below.

Comment (1) of Reviewer #1

As far as I understand, the authors estimated the THz electric field strength from the electro-optic sampling, as described in the Methods, and consequently calculated the corresponding intensity via the integral at the end of that section. From the calculated intensity, I guess that the authors estimated the corresponding THz energy of $\sim 10^{-13}$ J. I do not think that this very low energy is measurable, but it is obviously estimated. Whether estimated or measured, I hope that the authors mention explicitly in the manuscript how this magnitude was obtained. I further suggest numbering the equations in the Methods.

Reply to comment (1) of Reviewer #1

As Reviewer #1 pointed out, the energy of the emitted terahertz wave was so small that it was not directly measured. The details of its estimation method are described in Methods section. Considering the comment of Reviewer #1, we have revised the related description in the main paper as follows to make it clear that the energy of the emitted terahertz wave was estimated from the terahertz electric field waveform quantitatively measured.

[Line 20 to 21 of page 13 in the old MS]

“the energy of the terahertz radiation emitted from $[\text{Ni}(\text{chxn})_2\text{Br}]\text{Br}_2$ is 1.3×10^{-13} J (see Methods section)”

→

[Line 17 to 19 of page 14 in the revised MS]

“the energy of the terahertz radiation emitted from $[\text{Ni}(\text{chxn})_2\text{Br}]\text{Br}_2$ was evaluated to be 1.3×10^{-13} J from the electric-field waveform $E_{\text{THz}}(t)$ quantitatively measured (see Methods section)”

In addition, we numbered the equations in the Methods.

Comment (2) of Reviewer #1

I am curious to know the damage threshold of the used Mott insulator. This will additionally provide a fair comparison with the other materials investigated in this work. I think that a summary table listing the investigated materials and the various aspects of the comparison will be a good addition, but it is optional.

Reply to comment (2) of Reviewer #1

The bromine-bridged Ni-chain compound was damaged when the excitation fluences of the ω – and 2ω –pulses exceed 12.8 mJ/cm^2 and 0.95 mJ/cm^2 , respectively. The set of these excitation fluences was the damage threshold in the present experimental setup. In germanium, the damage threshold for the set of the excitation fluences of the ω – and 2ω –pulses was 25.5 mJ/cm^2 and 1.89 mJ/cm^2 , respectively. We showed these damage threshold values in Results section as follows.

[Line 12 to 22 of page 11 in the revised MS]

“When the terahertz radiation generated in this way is actually used to excite materials, the damage of each crystal when the excitation fluence is increased must also be considered. In the present experiment, when ω – and 2ω –pulses were irradiated simultaneously, the thresholds of the fluences of ω – and 2ω –pulses at which they cause damage of the crystal were 11.7 mJ/cm^2 and 0.86 mJ/cm^2 in $[\text{Ni}(\text{chxn})_2\text{Br}]\text{Br}_2$ (23.4 mJ/cm^2 and 1.74 mJ/cm^2 in Ge), respectively. The terahertz electric-field amplitude tends to saturate when the pulse fluences approach the damage thresholds in both materials. Just below the damage threshold, the terahertz electric-field amplitude was found to be 0.53 kV/cm in $[\text{Ni}(\text{chxn})_2\text{Br}]\text{Br}_2$ and 0.23 kV/cm in Ge. Considering these values, the ratio of their radiation intensities is decreased from 42 to 39, but the superiority of $[\text{Ni}(\text{chxn})_2\text{Br}]\text{Br}_2$ remains unchanged.”

In addition, the related sentence was modified as follows.

[Line 10 to 12 of page 10 in the old MS]

“Taking this factor into account, the terahertz radiation efficiency per unit length in $[\text{Ni}(\text{chxn})_2\text{Br}]\text{Br}_2$ is estimated to be 2.6×10^4 times as large as that of Ge.”

→

[Line 7 to 9 of page 12 in the revised MS]

“Taking into account this factor and the intensity ratio of 42 estimated above, the terahertz radiation efficiency per unit length in $[\text{Ni}(\text{chxn})_2\text{Br}]\text{Br}_2$ is estimated to be 2.6×10^4 times as large as that of Ge.”

Comment (3) of Reviewer #1

Regarding the matching of colors of the curves and legends of Fig. 5h, especially for the green and blue, I still do not think they really match, but I will leave this to be judged by the production team of Nature Communications.

Reply to comment (3) of Reviewer #1

Considering the comment of Reviewer #1, we have carefully chosen the colors of curves and legends in Fig. 5h so that they really match.

Comment of Reviewer #2

Adding new references, the authors revised the manuscript and it reads much better. However, the main message of the paper is unchanged. Again, I do not feel that the result represents a breakthrough or an important step in terahertz emission, light-matter interaction, or control of matter because essentially same phenomena have been reported in the literatures. The authors summarized the difference from previous studies as well-defined parities of excitons, high efficiency of terahertz radiation, and determination of phase relaxation time of excitons. But all the things look a minor extension. The high efficiency is a comparison among the methods using third-order nonlinearity, and the difference from conventional semiconductors is just a factor of 3 or 4. The obtained field is only 100 V/cm, which is much weaker than usual terahertz emission by typical second-order process. The authors expect a frequency tenability for the use of other compounds, but it is not demonstrated in this work.

I cannot recommend it for the publication in Nature Communication. I suggest Communication Physics as a better outlet for this work.

Reply to comment of Reviewer #2

We would like to appreciate his/her comment; “Adding new references, the authors revised the manuscript and it reads much better.”. We show the replies to his/her comments below, in which we divide his/her comments into 3 parts (comment a-c).

Comment (a) of Reviewer #2

Again, I do not feel that the result represents a breakthrough or an important step in terahertz emission, light-matter interaction, or control of matter because essentially same phenomena have been reported in the literatures. The authors summarized the difference from previous studies as well-defined parities of excitons, high efficiency of terahertz radiation, and determination of phase relaxation time of excitons. But all the things look a minor extension. The high efficiency is a comparison among the methods using third-order nonlinearity, and the difference from conventional semiconductors is just a factor of 3 or 4.

Reply to comment (a) of Reviewer #2

Reviewer #2 claims that the efficiency of the terahertz radiation from a 1D Mott insulator of the bromine-bridged Ni-chain compound is only three to four times higher than that from conventional semiconductors. This is a misunderstanding by Reviewer #2. As stated in line 15 on page 10 to line 16 on page 11 in the old MS, the terahertz radiation

efficiency per unit volume of the Ni-chain compound is very high, which is approximately 26,000 times higher than that of Ge. Our paper demonstrates that 1D Mott insulators have an important feature that they can generate terahertz radiations with much higher efficiency than conventional semiconductors.

Comment (b) of Reviewer #2

The obtained field is only 100 V/cm, which is much weaker than usual terahertz emission by typical second-order process.

Reply to comment (b) of Reviewer #2

In the experimental condition where the terahertz radiation was measured in our study, the terahertz electric-field amplitude generated from the Ni-chain compound is 107 V/cm. This amplitude is indeed small compared to that of a terahertz pulse commonly used as an excitation light, as commented by Reviewer #2. However, it is possible to enhance the electric-field amplitude of the terahertz pulse emitted from the Ni-chain compound by optimizing the experimental condition as described in (i)-(iii) below.

(i) The intensity of the femtosecond laser pulse used as the excitation of the Ni-chain compound to generate a terahertz pulse was very small, which is 1.8×10^{-6} J. In general, to generate a terahertz pulse with an electric-field amplitude of ~ 1 MV/cm, a femtosecond laser pulse of more than 1000 times the intensity is used as the excitation. In fact, in a previous study, a terahertz pulse of 1.2 MV/cm was generated by irradiating a nonlinear optical crystal with an intense femtosecond laser pulse of 4.0×10^{-3} J [H. Hirori *et al.* Appl. Phys. Lett. **98**, 091106 (2011)]. To enhance the intensity of the excitation laser pulse without damaging the sample, that is, the Ni-chain compound, the beam diameter of the excitation laser pulse should be broadened. When widening the beam diameter from 250 μm to 5 mm with the excitation photon density per unit area fixed, the intensity of the excitation laser pulse can be increased to 7.2×10^{-4} J. In this case, the intensity of the generated terahertz pulse would be increased in proportion to the excitation intensity, which is 400 times larger than that in the original case. This means that the terahertz electric-field amplitude would be increased by a factor of 20.

(ii) In our study, the terahertz pulse emitted from the Ni-chain compound was focused on the sample (and EO crystal), at which the beam diameter was approximately 1.1 mm. The electric field amplitude of the terahertz pulse applied on the sample can be enhanced by reducing the beam diameter. Roughly speaking, the diffraction limit of an electromagnetic wave is equal to its wavelength. In other words, a terahertz pulse with a frequency of 1

THz can be focused down to approximately 0.3 mm. If the beam diameter of the terahertz pulse is focused down to 0.3 mm, its electric-field amplitude can be increased by a factor of approximately 3.7.

(iii) In our experimental setup for the terahertz radiation measurements, the polarizations of the two types of pump lights (ω –pulse and 2ω –pulse) are tilted 45 degrees in opposite directions from the 1D Ni-Br chain direction to each other. As stated in the main paper, if the polarizations of both ω –pulse and 2ω –pulse are in the 1D Ni-Br chain direction, the electric-field amplitude of the generated terahertz pulse should be $2\sqrt{2}$ times higher.

By changing the experimental conditions as detailed in (i)-(iii), the terahertz electric-field amplitude will be able to be enhanced to 22 kV/cm. This is an electric-field amplitude that can be used as an excitation light. A phase-tunable terahertz pulse generated by the method reported in our paper will enable us to accelerate studies on the control of physical properties by a terahertz electromagnetic field. We also expect that we can obtain even higher electric-field amplitudes of terahertz pulses by searching for 1D Mott insulators suitable to our method. The explanations about the experimental conditions necessary to enhance the electric-field amplitude of the terahertz pulse mentioned above were somewhat complicated and not so important for general readers, therefore, we did not present them in the revised MS.

Comment (c) of Reviewer #2

The authors expect a frequency tenability for the use of other compounds, but it is not demonstrated in this work.

Reply to comment (c) of Reviewer #2

If the energies of the odd-parity and even-parity excitons in a 1D Mott insulator are far apart, it should be possible to generate a high-frequency terahertz pulse using the same technique as in this paper. However, in this case, femtosecond laser pulses with short temporal widths having spectral widths larger than the energy difference between two excitons must be used as the excitation pulses, and the photon energies of 2ω – and ω –pulses must be tuned to match the odd-parity exciton energy and half of the even-parity exciton energy, respectively. In addition, appropriate nonlinear optical crystals for the second harmonic generation and for the adjustment of time delay between 2ω – and ω –pulses are necessary. Thus, to search a good terahertz radiation material in the other

1D Mott insulators, the experimental setup needs to be significantly modified. We would like to leave this issue for future work.

Comment (0) of Reviewer #3

The manuscript deals with experimental demonstration of effective generation of terahertz waves from a typical 1D Mott insulator $[\text{Ni}(\text{chxn})_2\text{Br}]\text{Br}_2$. They claim to achieve this by exciting odd-parity, one-photon state and even-parity, two photon state using two-color femtosecond pulses and based on the quantum interference of the two third-order nonlinear optical responses associated with terahertz radiation, which are characterized by $\chi^{(3)}(-\omega_{\text{THz}}; \omega, \omega, -(2\omega - \omega_{\text{THz}}))$ and $\chi^{(3)}(-\omega_{\text{THz}}; -\omega, -\omega, 2\omega + \omega_{\text{THz}})$. The two nonlinear processes interfere, and the radiation intensity oscillates depending on the delay time of generation of the above mentioned two low energy excitations. The authors have discussed the physical mechanism of the observed terahertz radiation and the Δt dependence of its phase, amplitude, and frequency in the third-order optical nonlinearity framework. It was shown that the phase, amplitude, and frequency of the terahertz waves can be controlled by the fine tuning of the generation time difference of two excitons. They have estimated the energy conversion efficiency from two excitation pulses to a terahertz pulse in $[\text{Ni}(\text{chxn})_2\text{Br}]\text{Br}_2$ with their method using the third-order optical nonlinearity. The electric field amplitude of the terahertz radiation emitted from $[\text{Ni}(\text{chxn})_2\text{Br}]\text{Br}_2$ is found to be 5.4 times as large as that from ZnTe. In general, the finding of the work is very interesting and would be of interest to the large community of researchers working in theoretical, experimental aspects of non-linear optics of strongly correlated systems. Thus it may deserve publication. The current reviewer wants to draw attention to the authors on the following aspects.

Replies to comments (0) of Reviewer #3

We would like to appreciate his/her comment; “In general, the finding of the work is very interesting and would be of interest to the large community of researchers working in theoretical, experimental aspects of non-linear optics of strongly correlated systems. Thus it may deserve publication.”

We would like to reply to his/her comments (1) and (2) below. As for comment (2), we divided it into five sections from (2-a) to (2-e) and will respond to each of them.

Comment (1) of Reviewer #3

The way the manuscript is written it would appear to the readers till a large part of the manuscript that it is only a proposal and the authors are probably providing a fitting estimate(s). The authors should claim/mention in the introduction itself clearly that this work is an experimental demonstration of efficient terahertz generation for strongly

correlated material the physics of which is under active research of condensed matter physics community. In order to establish so what is achieved in which section/subsection must be mentioned at the end of the Introduction itself.

Reply to comment (1) of Reviewer #3

Following the suggestion of Reviewer #3, we have revised the sentence in the Introduction to make it clear that our study presents an experimental demonstration of efficient terahertz generation for strongly correlated materials and that the physics has been actively studied in the condensed matter physics community as follows.

[Line 7 to 9 of page 3 in the old MS]

“From these backgrounds, in this study, we propose an efficient generation method of nearly monocyclic terahertz pulses with adjustable phase, frequency and amplitude using excitonic states in one-dimensional (1D) Mott insulator.”

→

[Line 7 to 11 of page 3 in the revised MS]

“On the basis of these backgrounds, in this study, we experimentally demonstrate an efficient generation method of nearly monocyclic terahertz pulses with adjustable phase, frequency, and amplitude using excitonic states in one-dimensional (1D) Mott insulators, the linear and nonlinear optical properties of which have been actively studied in the community of condensed matter physics³⁵⁻⁴⁶.”

Comment (2-a) of Reviewer #3

Experimental findings may be correct but the physical explanations the authors provided and the model used for the support for explanation in the first few pages of the manuscript is questionable. It appears as if it is the authors who are first time proposing the terahertz generation in one dimensional strongly correlated quantum wire. This is not true (see below). In the process the authors ignore many important literatures and lose to attract general nature of their work in the subject.

Reply to comment (2-a) of Reviewer #3

The paper presented by Reviewer #3 does indeed theoretically predict that terahertz electromagnetic waves can be generated from 1D Mott insulators with high efficiency. Therefore, we agree with the suggestion of Reviewer #3 that those papers should be cited. On the other hand, the mechanisms of the terahertz radiation predicted in those studies are not the same as that reported in our paper but work only for materials in which the

even-parity excited state is slightly on the low energy side of the odd-parity excited state. The terahertz radiation mechanisms reported in those papers are summarized as follows.

In one of the papers Reviewer #3 introduced, it is theoretically predicted that for materials in which the even-parity excited state is slightly on the low energy side of the odd-parity excited state, an excitation of the odd-parity excited state will result in a terahertz radiation via its efficient relaxation to the even-parity excited state [H. Ghosh, *Europhysics Lett.* **75**, 468–474 (2006); H. Ghosh, *Phys. Rev. B* **75**, 235127 (2007)]. The terahertz radiation by this mechanism occurs via a spontaneous emission process. However, the phases of terahertz electromagnetic waves generated in those spontaneous emission processes are necessarily random. This means that the terahertz pulse thus generated cannot be used as an excitation pulse in the terahertz region.

Another paper Reviewer #3 introduced reported a theoretical prediction that the excitation of the odd-parity excited state may induce a terahertz electromagnetic wave with high efficiency as a Stokes radiation in the stimulated Raman scattering process [H. Ghosh and R. Chari, *Phys. Lett. A* **374**, 2379 (2010)]. Analogous to a Raman laser, the stimulated Raman scattering process may generate a coherent terahertz electromagnetic wave. However, if another terahertz pulse is not simultaneously injected as a seed light, a terahertz wave generated by a spontaneous Raman scattering process will be amplified and generated. In this case, the phase of the generated terahertz electromagnetic wave is random in each pulse and difficult to be controlled.

On the other hand, in our study, terahertz electromagnetic waves are generated based on a different principle. A 2ω –pulse is prepared directly from a ω –pulse using a second-harmonic generation process, and the odd-parity exciton and even-parity exciton in a 1D Mott insulator are generated by these 2ω –pulse and ω –pulse, respectively. In this condition, the coherences of the odd-parity and even-parity excitons can be maintained. This induces the quantum interference between the two excitons, which results in the generation of a terahertz electromagnetic wave with the frequency corresponding to the energy difference of the two excitons. In this method, by changing the generation time of the two excitons, we can control the phase difference between the odd-parity and even-parity excitons, that is, the phase of the radiated terahertz electromagnetic wave. Thus, our method is one step ahead of the methods previously proposed in that it can control the phase of the terahertz electromagnetic wave.

Considering the comment of Reviewer #3, we added the following sentence to the main paper showing that theoretical predictions for efficient terahertz radiations in 1D Mott insulators have been previously proposed and that the mechanism of the terahertz

radiation used in our study is different from those previously proposed.

[Line 6 to 9 of page 6 in the revised MS]

“Note that other methods for the efficient generation of terahertz electromagnetic wave via the third-order optical nonlinearity in 1D Mott insulators have been theoretically proposed^{42,44,45}, while in those methods the phase of the terahertz electromagnetic wave cannot be controlled unlike the method proposed in the present study.”

In addition, we have added the previous papers suggested by Reviewer #3 in Reference section as follows.

42. Ghosh, H. Giant ground- and excited-state optical nonlinearity in one-dimensional cuprates. *Europhysics Lett.* **75**, 468–474 (2006).
44. Ghosh, H. Excited state nonlinear optics of quasi-one-dimensional Mott-Hubbard insulators. *Phys. Rev. B* **75**, 235127 (2007).
45. Ghosh, H. & Chari, R. Gigantic stimulated Raman scattering in one-dimensional Mott-Hubbard insulators: A possible THz source. *Phys. Lett. A* **374**, 2379–2382 (2010).

Comment (2-b) of Reviewer #3

Gigantic optical nonlinearities are observed in various one dimensional Mott-Hubbard insulators represented by solid circles compared to other one dimensional materials (hollow circles). Experimental and theoretical research predicts occurrence of nearly degenerate one and two photon states in these one dimensional Mott-Hubbard insulators which are very strongly dipole coupled. This is the primary reason for huge ground state optical non-linearities like TPA, THG, EA. This story began in the year 2000 (Nature 405, 929–932 (2000), Physical Review Letters 85, 2204, (2000)).

Subsequently it was realized that not only the lowest one/two photon states are nearly degenerate but there would be similar higher energy two photon states in the electronic structure. `` Excited state nonlinear optics of quasi-one-dimensional Mott-Hubbard insulators”, Physical Review B 75, 235127, 2007 (and references therein). Based on exact numerical results on finite size one-dimensional Mott-Hubbard insulators as well as available experimental data it was already predicted to be a good source of terahertz radiation. Further predications with explicit calculations were published later in, “Gigantic stimulated Raman scattering in one-dimensional Mott–Hubbard insulators: A possible THz source” Physics Letters A 374, 2379 (2010); <https://doi.org/10.1016/j.physleta.2010.03.055> IF 2.654 ISSN 0375-9601. Both ground and excited state optical nonlinearities are required for efficient terahertz communication medium. “Giant ground and excited state optical non-linearity in one dimensional cuprates”, Europhysics Letters 75, 468 (2006); <https://doi.org/10.1209/epl/i2006-10337-8>. Question is, is the strong electron-electron repulsion (U) and/or confinement enough for terahertz communication source!? Then how to explain the material dependent properties presented in the figure! Does the holon-doublon model have enough! The reviewer requests authors’ attention to these facts! The smallest parameter in the game is the presence of “Br” atoms (its site energy, the electronegativity, note the orders of magnitude difference in $\text{Im}\chi^{(3)}$ in the figure even though they belong to the same class). Clearly, authors themselves were in doubt (/uncomfortable) in page 5, line 15.

The authors asserted that, “The intense terahertz pulse has been used as an excitation pulse to control the electronic state properties such as ..., superconductivity9–11, magnetism19–23...” It is well known that superconductivity is realized only with varied oxygen content (in cuprates); magnetism is also controlled through oxygen content. . Large J ~2800 K of Ni–Br–Br as compared with that of Ni–Cl–Cl and Ni–Cl–NO₃ is attributable to the smaller value of charge transfer gap (difference in site energies of Ni and X (= Br / Cl ...). The difference of J between Ni–Cl–Cl; J ~2200 K and Ni–Cl–NO₃ J~2100 K has the same origin) ; U or V is same or similar in all these compounds. (See for example, PHYSICAL REVIEW B 70, 085101 (2004)). There would be no exchange coupling without intervening legend atoms (Br, O etc.). In short, the absence of Br atoms in holon-doublon model is not a practical one as far as its application to non-linear optical medium required for terahertz communication is concerned. The authors use over simplified cartoon pictures which are arguable; For example, In Fig. 1a if the states |L> and |R> are separated from the ground state by U then so is their linear combinations Thus it cannot describe correctly the location (energetically) of the one/two- photon states in

Fig 2c. The difference in 1-/2-photon states in case of Sr2CuO3 is much smaller as compared to those in the Ni–Cl–Cl and Ni–Cl–NO3 but still Sr2CuO3 has much lower $\text{Im}\chi^{(3)}$ (cf. the figure). In case of Ca2CuO3 the energy orderings of the 1-ph/2-ph states are reversed. The holon-doublon model does not have enough ingredients to capture the essential physics of the whole classes of strongly correlated materials.

The appropriate model required for 1-d Mott-Hubbard insulators, in the present purpose is (Physics Letters A 374, 2379 (2010)),

$$H = t_{pd} \sum_{(ij),\sigma} (d_{i\sigma}^\dagger p_{j\sigma} + h.c.) + \varepsilon_d \sum_{i,\sigma} n_i^d - \varepsilon_p \sum_{i,\sigma} n_i^p + \sum_i U_{dd} n_{i\uparrow}^d n_{i\downarrow}^d + \sum_i U_{pp} n_{i\uparrow}^p n_{i\downarrow}^p + \sum_{i,\delta} V_\delta n_i n_{i+\delta}$$

The first term represents hopping probability between the Cu (Ni) and O (X) sites, the second and third terms represent respectively the on-site energies of the Cu (Ni) and O (X), thereby defining the charge transfer energy $2\varepsilon = \varepsilon_{\text{O(X)}} - \varepsilon_{\text{Cu(Ni)}}$. The fourth and the fifth terms represent respectively the onsite Coulomb repulsions, i.e., the repulsion energy felt by two electrons occupying the same Cu (Ni)-3d or O (X)-p orbital with opposite spins. The last term is known as intersite Coulomb repulsion (V), repulsive energy felt by two electrons occupying the neighboring Cu (Ni) and O (X) (second neighbor Cu (Ni) and Cu (Ni), etc.) sites. In fact these materials under considerations are more of charge transfer Mott insulators ... see for example, “Origin of Giant Optical Nonlinearity in Charge-Transfer–Mott Insulators: A New Paradigm for Nonlinear Optics”, Phys. Rev. Lett. 86, 2086 (2001). Furthermore indeed two band model is required to explain the observed large non-linearity and ultrafast recovery pre-requisite for THz communications; see for example, “Interband two-photon transition in Mott insulator as a new mechanism for ultrafast optical non-linearity”, Int. Jr. Mod. Phys. B15,3628(2001); <https://doi.org/10.1142/S0217979201008305> (also, Book Chapter on “Excitonic processes in condensed matter”, Edited by Kikuo Cho, Atsuo Matsui, World Scientific, ISBN 981-02-4588-2. Page no. 60 (2001)). Since the authors are NOT directly using the holon-doublon model for experimental demonstration of Thz radiation, they should use a practical model.

Reply to comment (2-b) of Reviewer #3

As reported previously, in a 1D Mott insulator of the bromine-bridged Ni-chain compound studied here, the odd-parity and even-parity excitons are almost degenerate, and the spatial extensions of their wavefunctions resemble each other except for their

phases. These features cause a large transition dipole moment between the two excitons, which is the main origin of the large third-order nonlinear optical response of this material. These features can be derived by a single-band Hubbard model [e.g., Y. Mizuno et al., *Phys. Rev. B* **62**, R4769–R4773 (2000); M. Ono *et al.*, *Phys. Rev. B* **70**, 085101 (2004)]. The mechanism of terahertz radiation reported in the present study can also be qualitatively explained using the single-band Hubbard model.

As Reviewer #3 pointed out, strictly speaking, the halogen-bridged Ni-chain compounds and the related materials of 1D cuprates are one-dimensional charge-transfer insulators. Therefore, to systematically explain the magnitudes of physical quantities such as energy gap, antiferromagnetic exchange interaction, and third-order nonlinear susceptibility in these materials, the two-band Hubbard model should be used. Considering the comment of Reviewer #3, we have revised the explanation of the terahertz radiation mechanism proposed in this study in the Introduction section which is based upon the two-band Hubbard model as follows.

[Line 13 of page 3 to line 16 of page 4 in the revised MS]

“In our study, we chose a bromine-bridged nickel-chain compound, $[\text{Ni}(\text{chxn})_2\text{Br}]\text{Br}_2$ (chxn=cyclohexanediamine), as a target material. In this compound, Ni^{3+} and Br^- ions are arranged alternately along the b -axis (Fig. 1a)⁴⁷. The singly occupied Ni- $3d_{z^2}$ orbitals, which are hybridized via the Br- $4p_z$ orbitals, form a 1D electronic band. The Ni- $3d$ band is split into the upper Hubbard band (UHB) and lower Hubbard band (LHB) due to the large U in the Ni- $3d_{z^2}$ orbital. Moreover, the Br- $4p$ band is located between the UHB and LHB, and the charge-transfer (CT) transition from the Br- $4p$ band to Ni- $3d$ UHB corresponds to the optical gap (Fig. 1b). Although this material is strictly a CT insulator described by the two-band Hubbard model considering Br- $4p$ and Ni- $3d_{z^2}$ orbitals^{39,48}, the single-band Hubbard model can explain the fundamental electronic properties of this material qualitatively^{36,38,40,41}. For this reason, we will simply call it a 1D Mott insulator in this paper.

In 1D Mott insulators, when the nearest-neighbor Coulomb repulsion energy is large, the low-energy electronic excited state is a doublon-holon bound state, i.e., an exciton⁴⁹. In this case, the lowest $|\varphi_o\rangle$ and second-lowest $|\varphi_e\rangle$ excitons are out of phase and in phase combination with an electron excitation from a Br ion to the left and right Ni ion

($|L\rangle$ and $|R\rangle$) (Fig. 1b), expressed as $|\varphi_o\rangle = \frac{(|L\rangle - |R\rangle)}{\sqrt{2}}$ and $|\varphi_e\rangle = \frac{(|L\rangle + |R\rangle)}{\sqrt{2}}$, respectively.

Thus, $|\varphi_o\rangle$ and $|\varphi_e\rangle$ are the odd-parity and even-parity excitons, respectively. If the hole in an exciton is located at the i site, the envelopes of the electron wavefunctions of $|\varphi_o\rangle$ and $|\varphi_e\rangle$ are as indicated by the blue and red lines in Fig. 1c,d, respectively. In this

case, their wavefunctions have similar spatial extensions except for their phases (Fig. 1c,d). Thus, their energy difference becomes minimal, and their transition dipole moment, $\langle \varphi_o | x | \varphi_e \rangle$, is enhanced^{35,36,40}. Owing to the large $\langle \varphi_o | x | \varphi_e \rangle$, 1D Mott insulators exhibit large third-order optical nonlinearity such as efficient third-harmonic generations (THG)^{38,41} and two-photon absorptions^{37,43}, and large electric-field changes of reflectivity^{35,40,46}. These features of excited states and large third-order optical nonlinearity in 1D Mott insulators were first explained by the single-band Hubbard model. Subsequently, calculations using the two-band Hubbard model have revealed that CT insulators indeed have similar features^{39,42,44,45}.

In addition, we have cited the related papers that considered the halogen-bridged Ni chain compounds as CT insulators and studied their electronic properties.

39. Zhang, G. P. Origin of Giant Optical Nonlinearity in Charge-Transfer–Mott Insulators: A New Paradigm for Nonlinear Optics. *Phys. Rev. Lett.* **86**, 2086–2089 (2001).
48. Okamoto, H. et al. Electronic structure of the quasi-one-dimensional halogen-bridged Ni complexes $[\text{Ni}(\text{chxn})_2\text{X}]\text{X}_2$ ($\text{X}=\text{Cl}, \text{Br}$) and related Ni compounds. *Phys. Rev. B* **54**, 8438–8445 (1996).

In the revised MS, we have also repositioned the following sentences for clarity of the explanations.

[Line 18 to 12 of page 5 in the old MS] → [Line 21 of page 5 to 2 of page 6 in the revised MS]

“Figure 2a shows the linear absorption spectrum, that is, the imaginary part of the dielectric constant, ε_2 , along the b -axis. It has a peak structure at 1.27 eV, which is attributed to the odd-parity exciton. Previous electro-reflectance spectroscopy results and THG studies revealed that the even-parity exciton is located at an energy level 10 meV (~ 2.5 THz) above the odd-parity exciton (Fig. 2a)⁴⁰. This satisfies the condition that $(\omega_e - \omega_o)$ is in the terahertz region.”

In connection with the above revisions, we have thoroughly revised Fig. 1, in which it is clearly shown that the bromine-bridged Ni-chain compound is a CT insulator. We have also deleted Fig. 2a in the old MS, which is not important.

Comment (2-c) of Reviewer #3

The used model in general perhaps cannot account for all the materials in the category of strongly correlated natural quantum wires; for example in case of Ca_2CuO_3 , the two photon state is below the one photon state. Possibly further, in that case $\chi^{(3)}(-\omega_{\text{THz}}; \omega, \omega, -(2\omega - \omega_{\text{THz}}))$ may not remain the dominant mechanism (authors should comment on that).

Reply to comment (2-c) of Reviewer #3

The energy of the lowest excited state with even parity in a 1D Mott insulator has been determined by applying the three-level model to the results of the electro-reflectance spectra [M. Ono et al. Phys. Rev. B **70**, 085101 (2004)]. In the analyses with the three-level model, the discrete energy levels of odd-parity and even-parity excitons are assumed as well as the ground state, and the third-order nonlinear susceptibility associated with those three states is considered. Therefore, in the three-level model, it is assumed that both of the lowest two excited states with odd-parity and even-parity are due to the excitons. In the Ni compound, $[\text{Ni}(\text{chxn})_2\text{Br}]\text{Br}_2$, reported here, it was revealed that the odd-parity and even-parity excitons are actually split off from the continuum, and the three-level model can reasonably explain the nonlinear optical responses experimentally obtained. On the other hand, in a 1D Mott insulator in which no excitons are stabilized and photo-excited states consist of continuous states, it was theoretically predicted that the dip structure in the spectrum of the electric-field induced absorption change ($\Delta\varepsilon_2$) appears on the lower energy side than the peak structure of the original absorption spectrum as shown in Figs. 3 and 4 in ref. 38 [H. Kishida *et al.* Phys. Rev. Lett. **87**, 177401 (2001)]. In this case, a spectrum analysis of the $\Delta\varepsilon_2$ spectrum using the three-level model gives a result as if an even-parity excited state exists on the lower energy side of an odd-parity excited state. The photoconductivity and ε_2 spectra revealed that the photoexcited state at the absorption edge in a 1D Mott insulator of Ca_2CuO_3 is not an exciton but a continuum [M. Ono et al. Phys. Rev. B **70**, 085101 (2004)]. From these facts, in Ca_2CuO_3 , the energy position of the even-parity excited state derived from the analysis of the $\Delta\varepsilon_2$ spectrum using the three-level model might not be exact. It is natural to consider that in the excited states of Ca_2CuO_3 , both the odd-parity and even-parity excited states would be continuous states and overlap each other in the spectrum. On the other hand, in two-dimensional Mott insulators of cuprates such as Nd_2CuO_4 and $\text{Sr}_2\text{CuO}_2\text{Cl}_2$, it has been revealed that the even-parity excitonic state is on the lower energy side than the odd-parity excitonic state [T. Terashige *et al.* Sci. Adv. **5**, eaav2187 (2019)]. In such systems, terahertz radiation may occur by a mechanism different from that reported in

this paper. However, when even-parity excitons are excited first in $[\text{Ni}(\text{chxn})_2\text{Br}]\text{Br}_2$, where the even-parity exciton has the higher energy than the odd-parity exciton, $\chi^{(3)}(-\omega_{\text{THz}}; \omega, \omega, -(2\omega - \omega_{\text{THz}}))$ dominates the terahertz radiation mechanism as we detailed in this paper.

In this way, in Ca_2CuO_3 , it is an interesting question to explore what kind of terahertz radiation occurs when our method is applied, as excitons are not stable in this material. Furthermore, systematically detecting terahertz radiation in Mott insulators, including two-dimensional systems, is highly intriguing. These are all topics that we would like to address as future works.

These discussions will be somewhat complicated for general readers. So, we have not added them in the revised MS.

Comment (2-d) of Reviewer #3

It is well known that the low energy excitations in 1-d strongly correlated systems are gapless spin excitations. Thus, authors should provide an estimate of “the delay of the generation time of the odd-parity excitons relative to the even-parity excitons” in contrast to the decay time to the ground state.

Reply to comment (2-d) of Reviewer #3

In the present study, the ω –pulse is used to generate even-parity excitons first, but no terahertz electromagnetic waves are generated via their relaxations to odd-parity excitons. In fact, when the 2ω –pulse is used to generate the odd-parity excitons while the even-parity excitons exist preserving their coherence, terahertz waves are emitted due to the quantum interference between the two types of exciton states. “The delay in the generation time of the odd-parity exciton relative to the even-parity exciton” in the comment of Reviewer #3 is not the relaxation time from the even-parity to the odd-parity exciton but the time delay of the 2ω –pulse relative to the ω –pulse. If the time delay is sufficiently large, the phases of the excitons initially excited are disturbed before the other excitons are generated. In this case, the quantum interference effect between the two types of excitons does not occur. In this context, it is important to compare the phase relaxation time of excitons with the relaxation time of excitons to the ground state. In another 1D CT insulator of $[\text{Ni}(\text{C14-en})_2\text{Br}]\text{Br}_2$ (C14-en=1,2-diaminohexadecane), the relaxation time of the lowest odd-parity excitons to the ground state was evaluated to be approximately 1.4 ps [T. Miyagoe *et al.* J. Phys. Soc Jpn. **77**, 023711 (2008)]. Although this value is for the odd-parity excitons, the relaxation time of the even-parity excitons in

[Ni(chxn)₂Br]Br₂ is expected to have a similar time scale. This is sufficiently long compared to the phase relaxation time (38 fs) of the even-parity excitons estimated in this study and therefore the effect of the relaxation processes to the ground state on the time characteristic of the terahertz radiation phenomenon can be neglected.

To clearly state the fact that the relaxation time of excitons to the ground state is sufficiently long compared to their phase relaxation time of excitons, we modified the related descriptions as follows.

[Line 22 of page 16 to line 1 of page 17 in the old MS]

“The non-oscillatory component $b(\Delta t)$ is characterized by the phase relaxation time of the even-parity exciton, $\tau_{\text{even}} = 38$ fs, which is much shorter than the exciton phase relaxation times in inorganic semiconductors, which are typically 10 ps⁵⁶. ”

→

[Line 20 of page 17 to line 1 of page 18 in the revised MS]

“The non-oscillatory component $b(\Delta t)$ is characterized by the phase relaxation time of the even-parity exciton, $\tau_{\text{even}} = 38$ fs, which is sufficiently short compared to its relaxation time to the ground state of approximately 1 ps⁶¹. This means that the effect of the relaxation to the ground state of the even-parity exciton on the time characteristic of $b(\Delta t)$ can be neglected. Furthermore, τ_{even} is much shorter than the exciton phase relaxation times in inorganic semiconductors, which are typically 10 ps⁶². ”

We have also added the related reference shown below.

61. Miyagoe, T. et al. Ultrafast Optical Responses in a One-Dimensional Mott Insulator of a Br-Bridged Ni Compound. *J. Phys. Soc. Jpn.* **77**, 023711 (2008).

Comment (2-e) of Reviewer #3

The gapless spin excited states are very similar in nature to the Antiferromagnetic ground state and being gapless it is expected that some of the population generation will also take place from these states. Authors should comment on such and other competitive processes.

Reply to comment (2-e) of Reviewer #3

The band dispersion of spin excitations in one-dimensional antiferromagnetic Heisenberg spin chains, such as 1D Mott insulators discussed here, is indeed zero gap. On the other hand, the maximum energy of the spin excitation in the dispersion is known to be $\pi J/2$ [Fig. 2(b) in H. Suzuura *et al.*, *Phys. Rev. Lett.* **76**, 2579 (1996)]. Since the

magnitude of the antiferromagnetic exchange interaction in the bromine-bridged Ni complex is $J=2800$ K [M. Ono et al. Phys. Rev. B **70**, 085101 (2004)], $\pi J/2 = 4400$ K, which is very high compared to the measured temperature (294 K). Therefore, it is reasonable to assume that only low-wavenumber spin excitations are thermally excited and that the ground state basically remains in the antiferromagnetic order.

In addition, 1D Mott insulators with large U have the property that charge carriers are not affected by spin degrees of freedom in their motion [M. Ogata and H. Shiba, Phys. Rev. B **41**, 2326 (1990)]. This property is called spin-charge separation. Because of this property, the effect of spin-excited states can be considered separately from exciton production. Indeed, it was revealed that the spectral width of the lowest excitonic transition is determined by the interaction with optical phonons and is not affected by spin degrees of freedom [M. Ono *et al.* Phys. Rev. B **70**, 085101 (2004)]. Moreover, it was demonstrated that the optical conductivity spectrum due to the odd-parity exciton and continuum calculated using the spinless charge model is almost perfectly the same as that calculated using the Hubbard model with including the spin degrees of freedom [S. Ohmura et al. Phys. Rev. B **100**, 235134 (2019)]. This means that the photoexcited states of 1D Mott insulators are largely unaffected by spin degrees of freedom. On the basis of those previous studies, we do not think that the excitonic states are strongly affected by the spin excitations. Therefore, we consider that the effect of the spin excitation on the terahertz radiation phenomenon originating from the interference of excitonic states is negligibly small.

We believe that the above discussions are somewhat complicated and not so important for the general reader. Therefore, we did not add them in the revised MS.

REVIEWERS' COMMENTS

Reviewer #3 (Remarks to the Author):

The revised manuscript is reasonably improved. The authors have adequately addressed all the questions raised by all the reviewer and incorporated it into the revised manuscript. The current reviewer has gone through all the responses corresponding to all the reviewers and reasonably satisfied with the authors responses. The only weak link that still remains to the current reviewer is that the mechanism of Terahertz radiation by quantum interference of excitons in these classes of one-dimensional Mott insulators remain same or not. This will not be apparent from this manuscript and it looks specific to the Br-compound. Thus my humble request to the authors would be to make a short discussion on the above and further challenges.

Leaving the above to the authors and editors, it is personal opinion of the reviewer that the rebuttal of the manuscript describes commendable progress in the practical application aspects of non-linear optics of strongly correlated systems based on essential difference on conventional sources. In general, the finding of the work would attract larger community of researchers working in theoretical, experimental condensed matter Physics as well as to those working on aspects of non-linear optics of strongly correlated systems. Thus it may deserve publication and may be published in Nature communication.

Summary of changes

(1)

Taking comment (1) of Reviewer #3 into account, we have added the explanation about the effectivity of the terahertz radiation mechanism presented here in the other 1D Mott insulators.

(2)

In line with the guideline described in the author checklist, the final paragraph of the Introduction section has been slightly amended to start with “in this work”.

[Line 10 to 13 of page 6 in the revised MS]

“In this work, we demonstrate that terahertz electromagnetic waves are generated via the quantum interference between the odd-parity and even-parity excitons in $[\text{Ni}(\text{chxn})_2\text{Br}]\text{Br}_2$, which are excited via one-photon and two-photon absorption processes, respectively, using two-color femtosecond laser pulses. The amplitude....”

(3)

In line with the guideline described in the author checklist, we have defined the symbols in all the figures in the associated legends.

(4)

We have corrected the Data Availability statement.

(5)

In line with the guideline described in the author checklist, we have reordered the sections as specified.

(6)

In line with the guideline described in the author checklist, we have revised the labels of equations in the Supplementary Information.

All the changed parts are marked in yellow in Marked MS3.pdf and Marked SI3. pdf.

Reply to comments of Reviewer #3

Comment (1) of Reviewer #3

The revised manuscript is reasonably improved. The authors have adequately addressed all the questions raised by all the reviewer and incorporated it into the revised manuscript. The current reviewer has gone through all the responses corresponding to all the reviewers and reasonably satisfied with the authors responses. The only weak link that still remains to the current reviewer is that the mechanism of Terahertz radiation by quantum interference of excitons in these classes of one-dimensional Mott insulators remain same or not. This will not be apparent from this manuscript and it looks specific to the Br-compound. Thus my humble request to the authors would be to make a short discussion on the above and further challenges.

Reply to comment (1) of Reviewer #3

We would like to thank Reviewer #3 for the thoughtful comment. We expect that the terahertz radiation mechanism can work in the other 1D Mott insulators. This point is briefly discussed in the Discussion section of the previous MS as shown below. Taking this comment of Reviewer #3 into account, we have added some explanations in the relevant part as follows.

[Line 19 of page 16 to line 2 of page 17 in the previous MS]

“In the series of halogen-bridged nickel-chain compounds expressed as $[\text{Ni}(\text{chxn})_2\text{X}]Y_2$, the energy difference between the even-parity and odd-parity excitons, $\hbar(\omega_e - \omega_o)$, can be widely changed from 10 meV to 90 meV by the choice of bridging halogen ion X and counterion Y^{40} . Therefore, if we appropriately select a temporal width of the incident laser pulses, a material with the desired energy of $\hbar(\omega_e - \omega_o)$, and a frequency range of the terahertz radiation detectable, it would be possible to control the terahertz radiation frequency over a wider range by varying Δt .”

→

[Line 19 of page 16 to line 9 of page 17 in the revised MS]

“In the series of halogen-bridged nickel-chain compounds expressed as $[\text{Ni}(\text{chxn})_2\text{X}]Y_2$, the energy difference between the even-parity and odd-parity excitons, $\hbar(\omega_e - \omega_o)$, can be widely changed from ~10 meV in $[\text{Ni}(\text{chxn})_2\text{Br}]\text{Br}_2$ to ~90 meV by the choice of bridging halogen ion X and counterion Y^{40} . When the energies of the odd-parity and even-parity excitons in a 1D Mott insulator are far apart, e.g., $\hbar(\omega_e - \omega_o) \sim 90$ meV, it will be possible to generate a high-frequency terahertz pulse with the frequency of ~20 THz using

the same technique. In that case, femtosecond laser pulses with short temporal widths having spectral widths larger than ~ 90 meV must be used as the excitation pulses, and the photon energies of $2\omega -$ and $\omega -$ pulses must be tuned to match the odd-parity exciton energy $\hbar\omega_o$ and half of the even-parity exciton energy, $\hbar\omega_e/2$, respectively. In addition, appropriate nonlinear optical crystals for the second harmonic generation and for the adjustment of time delay between $2\omega -$ and $\omega -$ pulses, Δt , are necessary. If these conditions are appropriately prepared, it may be possible to achieve the high-frequency terahertz pulse generation and to control the frequency over a wider range by varying Δt . This is an interesting issue in future.”

Comment (2) of Reviewer #3

Leaving the above to the authors and editors, it is personal opinion of the reviewer that the rebuttal of the manuscript describes commendable progress in the practical application aspects of non-linear optics of strongly correlated systems based on essential difference on conventional sources. In general, the finding of the work would attract larger community of researchers working in theoretical, experimental condensed matter Physics as well as to those working on aspects of non-linear optics of strongly correlated systems. Thus it may deserve publication and may be published in Nature communication.

Reply to comment (2) of Reviewer #3

We would like to thank Reviewer #3 for the supportive comment.